# A cold laboratory hyperspectral imaging system to map grain size and ice layer distributions in firn cores

Ian E. McDowell[1,2], Kaitlin M. Keegan[2], S. McKenzie Skiles[3], Christopher P. Donahue[4], Erich C. Osterberg[5], Robert L. Hawley[5], and Hans-Peter Marshall[6]

[1]Graduate Program of Hydrologic Sciences, University of Nevada, Reno, Reno, NV, USA
[2]Department of Geological Sciences and Engineering, University of Nevada, Reno, Reno, NV, USA
[3]Department of Geography, University of Utah, Salt Lake City, UT, USA
[4]Department of Geography, Earth and Environmental Sciences, University of Northern British Columbia, Prince George, BC, Canada
[5]Department of Earth Sciences, Dartmouth College, Hanover, NH, USA
[6]Department of Geosciences, Boise State University, Boise, ID, USA

**Correspondence:** Ian E. McDowell (imcdowell@unr.edu)

**Abstract.** The Greenland and Antarctic ice sheets are covered in a layer of porous firn. Knowledge of firn structure improves our understanding of ice sheet mass balance, supra- and englacial hydrology, and ice core paleoclimate records. While macroscale firn properties, such as firn density, are relatively easy to measure in the field or lab, more intensive measurements of microstructural properties are necessary to reduce uncertainty in remote sensing observations of mass balance, model meltwater infiltration, and constrain ice age – gas age differences in ice cores. Additionally, as the duration and extent of surface melting increases, refreezing meltwater will greatly alter firn structure. Field observations of firn grain size and ice layer stratigraphy are required to test and validate physical models that simulate the ice sheet-wide evolution of the firn layer. However, visually measuring grain size and ice layer distributions is tedious, time-consuming, and can be subjective depending on the method. Here we demonstrate a method to systematically map firn core grain size and ice layer stratigraphy using a near-infrared hyperspectral imager (NIR-HSI; 900-1700 nm). We scanned 14 firn cores spanning ∼1000 km across western Greenland's percolation zone with the NIR-HSI mounted on a linear translation stage in a cold laboratory. We leverage the relationship between effective grain size, a measure of NIR light absorption by firn grains, and NIR reflectance to produce high-resolution (0.4 mm) maps of effective grain size and ice layer stratigraphy. We show the NIR-HSI reproduces visually-identified ice layer stratigraphy and infiltration ice content across all cores. Effective grain sizes change synchronously with traditionally-measured grain radii with depth, although effective grains in each core are 1.5x larger on average, which is largely related to the differences in measurement techniques. To demonstrate the utility of the firn stratigraphic maps produced by the NIR-HSI, we track the 2012 melt event across the transect and assess its impact on deep firn structure by quantifying changes to infiltration ice content and grain size. These results indicate that NIR-HSI firn core analysis is a robust technique that can document deep and long-lasting changes to the firn column from meltwater percolation, while quickly and accurately providing detailed firn stratigraphy datasets necessary for firn research applications.

# 1 Introduction

Many important glaciological research applications, such as interpreting previous atmospheric compositions using ice cores, monitoring ice sheet mass balance through remote sensing, and quantifying the firn's capacity to buffer future sea level rise by storing meltwater, rely on an understanding of firn structure. Firn can be defined in multiple ways depending on the field of study or glaciological environment. In ice sheet settings subject to surface melting, firn is defined as snow that remains after an ablation season but that has not been transformed into glacial ice (Cogley et al., 2011; The Firn Symposium Team, 2024). Structurally, firn is the intermediate metamorphic stage between fresh snow and glacial ice, containing at least some interconnected pore space with typical densities ranging between 400–830 kg m$^{-3}$ (Cogley et al., 2011). Spatially-extensive accumulation zones result in firn covering approximately 90% of the Greenland Ice Sheet and 99% of the Antarctic Ice Sheet surfaces (Noël et al., 2022; Winther et al., 2001). The firn volume of ice sheets is abundant; depending on site temperature and accumulation rate, the maximum firn column thickness can range between $\sim$ 40 m (Hollin and Cameron, 1961) and 120 m (Ligtenberg et al., 2011). The interconnected interstitial spaces between firn grains, i.e., open porosity, allows for gas, vapor, and liquid movement within the column; however, the total open porosity of the firn column is dependent on local climate conditions (e.g., Gregory et al., 2014) and can be progressively reduced by filling with meltwater (e.g., Harper et al., 2012). Therefore, an understanding of firn structure and properties, and their spatiotemporal evolution, is critical to determine how ice sheets respond to changes in climate.

Macroscale firn properties, such as density or porosity, are relatively easy to obtain in field or laboratory settings and can be used to determine the depth to pore close-off, where firn transitions to glacial ice and is impermeable to air flow (e.g., Schwander and Stauffer, 1984; Westhoff et al., 2023), ascertain past accumulation rates from ice-penetrating radar measurements (e.g., Miège et al., 2013; Hawley et al., 2014; Lewis et al., 2019), and estimate the total capacity for meltwater storage in the firn layer by integrating porosity over the firn depth (e.g., Harper et al., 2012; Vandecrux et al., 2019). However, density has been shown to correlate poorly with firn permeability, which controls the movement of fluid through firn (e.g., Adolph and Albert, 2014; Gregory et al., 2014; McDowell et al., 2020), indicating that density is not the only metric that should be used to characterize firn structure.

Microstructural properties, such as grain size, have been shown in previous studies to be necessary for improving our understanding of firn structure evolution. The relationship between gas diffusivity and firn permeability differs depending on firn grain size (Adolph and Albert, 2014) and pore close-off in finer-grained firn layers is reached at shallower depths in the firn column than it is for coarser-grained layers regardless of the density of the layers at depth (Gregory et al., 2014). These grain size effects must be accounted for when determining ice age – gas age differences in ice core records. Additionally, analysis of microwave and optical remotely-sensed measurements to determine ice sheet mass balance requires consideration of firn grain size. The firn's grain size controls the penetration depth of high-frequency (>10 GHz) microwaves in firn and governs scattering and emissivity (Rott et al., 1993; Brucker et al., 2010). Lastly, firn grain size regulates meltwater flow in firn by controlling capillary forces and water-entry pressures that need to be satisfied before water can percolate into unsaturated firn layers (Katsushima et al., 2013). Grain size transitions between adjacent firn layers can create capillary barriers that stall

vertical meltwater infiltration (e.g., Marsh and Woo, 1984; Eiriksson et al., 2013; Avanzi et al., 2016), which hinder meltwater from reaching deep pore space, especially if ice layers form as meltwater refreezes (McDowell et al., 2023) and will reduce the overall meltwater storage capacity in firn. Therefore, grain size data are crucial for improving our understanding of firn structure and reducing uncertainty in ice age – gas age differences, remotely-sensed mass balance changes, and meltwater fate and transport in firn.

Ice layers that freeze within the firn column will further complicate interpretations of ice cores, mass balance assessments from microwave radar surveys, and estimates of the firn's meltwater retention capacity. As surface melt events increase in duration and extent in Greenland (Colosio et al., 2021), firn structure will progressively be modified by ice layers across the ice sheet (MacFerrin et al., 2019; Culberg et al., 2021). Ice layers reduce vertical firn permeability, which alters gas transport dynamics and can reduce confidence in the accuracy of climate reconstructions (Keegan et al., 2014; Sommers et al., 2017), while also allowing meltwater to refreeze before accessing deep pore space. Satellite altimetry interpretations are also made more challenging by ice layer formation, since ice layers form strong radar reflectors and change the firn's scattering properties (Nilsson et al., 2015; Simonsen and Sørensen, 2017). Additionally, the low-permeability horizons created by ice layers allow for thick, impermeable ice slabs to amalgamate and render deep pore space inaccessible to meltwater, which expands the runoff zone into Greenland's interior and reduces total meltwater storage capacity (Machguth et al., 2016; MacFerrin et al., 2019; Culberg et al., 2021). Our ability to describe each of these processes will require the capability to measure the extent of ice layers in firn, especially given that their areal extent will increase under future climate scenarios (MacFerrin et al., 2019).

Datasets containing information on firn microstructure and ice layer stratigraphy are necessary given their importance to firn applications. Unfortunately, grain size measurements of firn are limited given the difficulty and time required to obtain them. Firn grain size datasets include "traditional" measurements produced by measuring the largest extent of grains using either a crystal card (e.g., Harper and Bradford, 2003), thin sections (e.g., Gow, 1969; Alley et al., 1982), or digital photographs (e.g., McDowell et al., 2023); outlining grain boundaries in scanning electron microscope (SEM) scans (Spaulding et al., 2010); extracting grain or crystal boundaries by tracing thermal grooves from optical microscope images of sublimed microtomed thin/thick sections (e.g., Kipfstuhl et al., 2009; Stoll et al., 2021); or calculating the specific surface area in microcomputer tomography (microCT) measurements (e.g., Freitag et al., 2004; Linow et al., 2012). While these methods are time-consuming and tedious, they include additional downsides: measuring traditional grain extents visually can be subjective (e.g., Baunach et al., 2001; Leppänen et al., 2015), while sample preparation for microCT, SEM, and optical microscope analyses is destructive to existing cores and their small size limits their representativeness. Grain size estimates produced using these techniques are averaged over the sample depth to obtain characteristic statistics and therefore do not produce continuous grain size profiles. Additionally, augmenting these records with ice layer stratigraphy requires visually inspecting firn cores or snowpit walls. These disadvantages motivate the development of a method that can quickly, continuously, and systematically map firn grain size and ice layer stratigraphy.

Spectroscopic studies of snow and ice provide an avenue to determine both firn grain size and ice layer distributions. Ice increasingly absorbs near-infrared (NIR) wavelengths (800 – 2500 nm), which produces a characteristic decline in the reflectance spectra of snow grains over these wavelengths. Techniques leveraging absorption mechanisms of ice can produce estimates of

the effective grain size. Effective grain size is commonly represented by the effective grain radius ($r_e$), which refers to the area-weighted radius of a sphere with the same surface area-to-volume ratio (specific surface area) as a snow grain (Wiscombe and Warren, 1980) and is therefore different from a traditional definition of grain size. The surface area-to-volume ratio of a sphere allows for a simple calculation to convert between $r_e$ and specific surface area, since the ratio is equal to $3/r_e$. Snow grain size has been determined by matching the theoretical reflectance spectra generated by modeling the hemispheric reflectance from a collection of effective spherical snow grains with the observed reflectance spectra from the snow sample (Grenfell and Warren, 1999). Additionally, since ice strongly absorbs particular wavelengths of NIR radiation, the size of these absorption features in the reflectance spectra can be inverted using modeled reflectance spectra to retrieve grain size (e.g., Nolin and Dozier, 1993, 2000). The millimeter – centimeter pixel resolution of near-infrared hyperspectral imagers (NIR-HSI) produces grain size maps at a high spatial and spectral resolution. Studies utilizing a NIR-HSI have been proven to efficiently and accurately produce high-resolution maps of grain size of laboratory snow samples (Donahue et al., 2021), along the vertical profile of a snowpit wall (Donahue et al., 2022), and at the snow surface when mounted to a drone (Skiles et al., 2023). In addition to providing grain size data, regions of low reflectance in the high-resolution images can allow for ice layers to be easily detected (Donahue et al., 2021).

This study was motivated by the need for high-resolution datasets of firn grain size and ice layer stratigraphy for a variety of firn research applications. We aimed to test the performance of a NIR-HSI system in retrieving accurate and continuous grain size profiles and ice layer distributions from 14 firn cores in a cold laboratory. To evaluate the efficacy of the NIR-HSI grain size retrievals, we (1) tested the sensitivity of retrieved effective grain sizes to the orientation of firn cores and the objective lens focus of the NIR-HSI; (2) compared the effective grain size retrievals with "traditional" grain size measurements colocated in 7 cores; and (3) correlated visual ice layer distributions with ice layer stratigraphy generated by the NIR-HSI. We establish that scanning firn cores with a NIR-HSI is a robust technique for developing detailed grain size and ice layer profiles, and demonstrate an application of the high-resolution dataset to quantify structural changes to the firn column following the extreme 2012 summer melt event.

## 2 Methods

### 2.1 Firn cores

We scanned 14 firn cores collected during the 2016 – 2017 Greenland Traverse for Accumulation and Climate Studies (Green-TrACS) (Graeter et al., 2018; Lewis et al., 2019). GreenTrACS consisted of two summer snowmobile traverses that approximately followed the 2200 m above sea level elevation contour. Firn cores were spaced approximately 40–100 km apart. Cores 1–7 were collected during the 2016 traverse that started at Raven-Dye-2 and ended at Summit, while cores 8–16 were collected during the ∼ 1200 km counter-clockwise loop that began and ended at Summit in 2017 (Figure 1). All cores were collected using a hand auger with a sidewinder attachment and reached depths between 20 – 30 m.

While collected outside of firn aquifer regions (Miège et al., 2016) and zones with thick ice slabs (MacFerrin et al., 2019), cores were drilled in locations targeted to preserve ice layers to use as a paleoclimate proxy resolving changes in annual surface

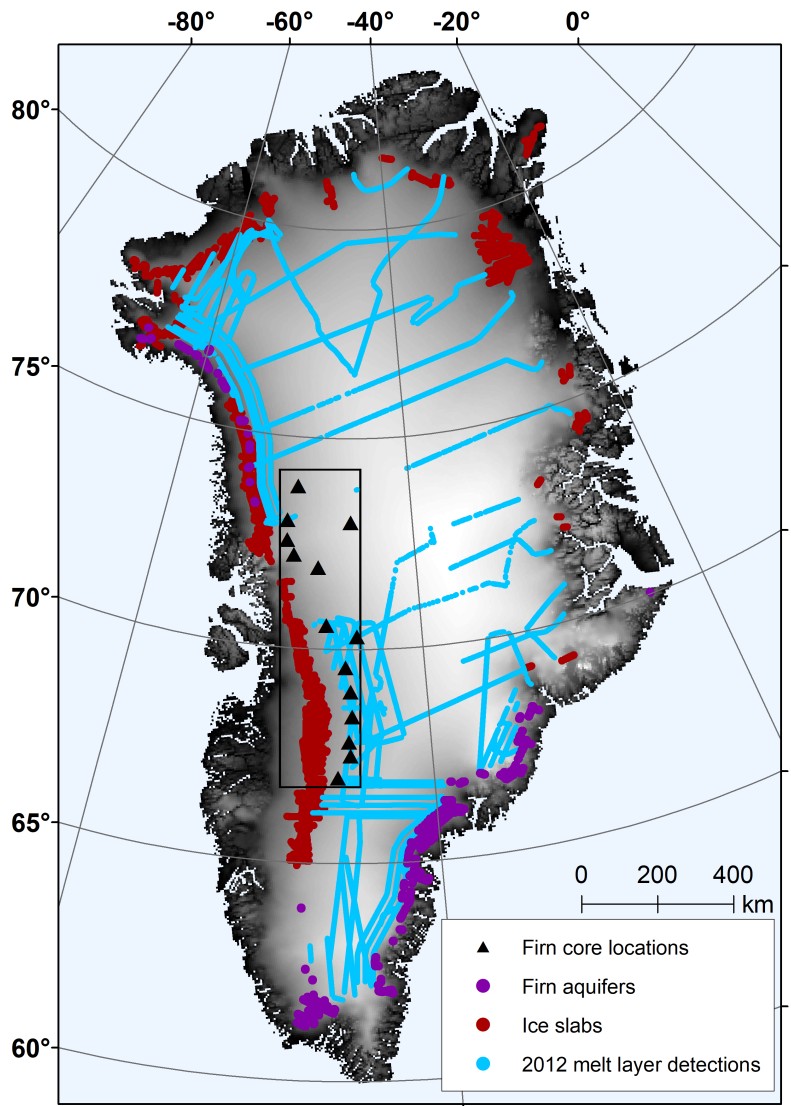

**Figure 1.** Locations of GreenTrACS cores in relationship to known firn aquifer locations (Miège et al., 2016), ice slab detections (MacFerrin et al., 2019), and the 2012 melt layer (Culberg et al., 2021). The outlined black box indicates the inset extent in Figure 5.

melt intensity (Graeter et al., 2018). Airborne radar reflectors indicated that most cores likely contained ice layers formed during the intense 2012 summer melt event (Culberg et al., 2021; Figure 1). Firn cores began at the base of snowpits ∼1 m deep, since

unconsolidated surface snow is difficult to drill and transport. The cores reached depths of approximately 20–30 m and the mass, diameter, and lengths of 0.03–1 m segments were measured in the field and again in the cold laboratory at Dartmouth College to calculate density (Graeter et al., 2018; Lewis et al., 2019). After cores were transported to the Dartmouth College ice core freezer via a commercial freezer truck, they were cut into half-round sections and sampled for chemical analysis of

water isotopes and major ion concentrations using a continuous melting system with discrete sampling (Osterberg et al., 2006; Graeter et al., 2018). Depth-age scales were generated for each core by identifying robust seasonal variations of $\delta^{18}O$ and other geochemical methods consistent with previous ice core studies (Graeter et al., 2018). The half-round cores were carefully inspected on a light table to develop a visual record of ice layer stratigraphy, and Cores 1–7 were photographed to generate a dataset of traditional grain size measurements (McDowell et al., 2023). We utilized the top 10 m of 14 cores for this study, since Cores 9 and 15 had deteriorated or broken during transport and storage.

## 2.2  Instrumentation and laboratory setup

We used a Resonon Inc. Pika NIR-320 NIR-HSI to scan firn cores in the cold laboratories at Dartmouth College and the University of Nevada, Reno. Donahue et al. (2021) provide a complete description of the instrument that we briefly describe here. The NIR-HSI measures 164 channels across the spectral range 900 - 1700 nm, resulting in a $\sim 4.9$ nm spectral resolution. The imager operates as a line scanner, or push-broom scanner, so it needs to either translate or rotate relative to the target scene, or the scene needs to translate relative to the stationary imager in order to generate a 2-D image with the full reflectance spectra in each pixel.

To collect images in the cold laboratory we mounted the imager to a commercially-available benchtop scanning stage that consisted of a linear stabilization rail and a motorized sliding platform that translated the imager across a stationary firn core. The benchtop scanning stage had a load capacity of 20 kg and is commonly used to mount cameras for time-lapse photography/videography. The rail was mounted above a table that held firn core sections either by using two tripods or by c-clamping the rail onto a metal bar above the table (Figure 2). We positioned the imager lens approximately 50 cm directly above the firn core, and the imager translated from the top to the bottom of the firn core segment during each scan. Because the firn cores had been cut into half-rounds to collect chemistry measurements, the flat surface of the core was leveled on the table and faced the imager. The firn core was illuminated by two 500 W halogen lamps that were positioned so that the core was evenly illuminated and the lights were no more than 5 ° off nadir. While these light sources were kept as close to vertical as possible, they could not be be directly above the scanning stage because it would cast a shadow on the core during the scan. However, Donahue et al. (2021) demonstrated that at a nadir viewing angle, illumination angle variability ranging between $0°$ $– 5°$ off nadir does not significantly impact grain radius retrievals. To prevent the firn cores from being warmed by the heat emitted from the halogen lamps, they were only turned on for the duration of each scan, which lasted $\sim$10 seconds. Figure 2 shows both a photograph and a diagram of the laboratory setup.

Focusing the line-scanning imager was the most time-consuming portion of the scanning process, since the imager had to be removed from the cold laboratory and stored when not actively in use. We focused the objective lens and determined the appropriate scan speed by moving the imager over the focus and calibration sheet provided by Resonon. The sheet consisted of a series of concentric thin black circles on a white background. The lens was adjusted until the outlines appeared focused, and we adjusted the speed of the stage until the rings appeared circular in the scanned image. If the imager moved too quickly over the target, the rings appeared vertically elongated, while if the scanner moved too slowly the rings appeared horizontally stretched. Once we determined the optimal scanning speed for our laboratory setup it was kept constant for all scans. However,

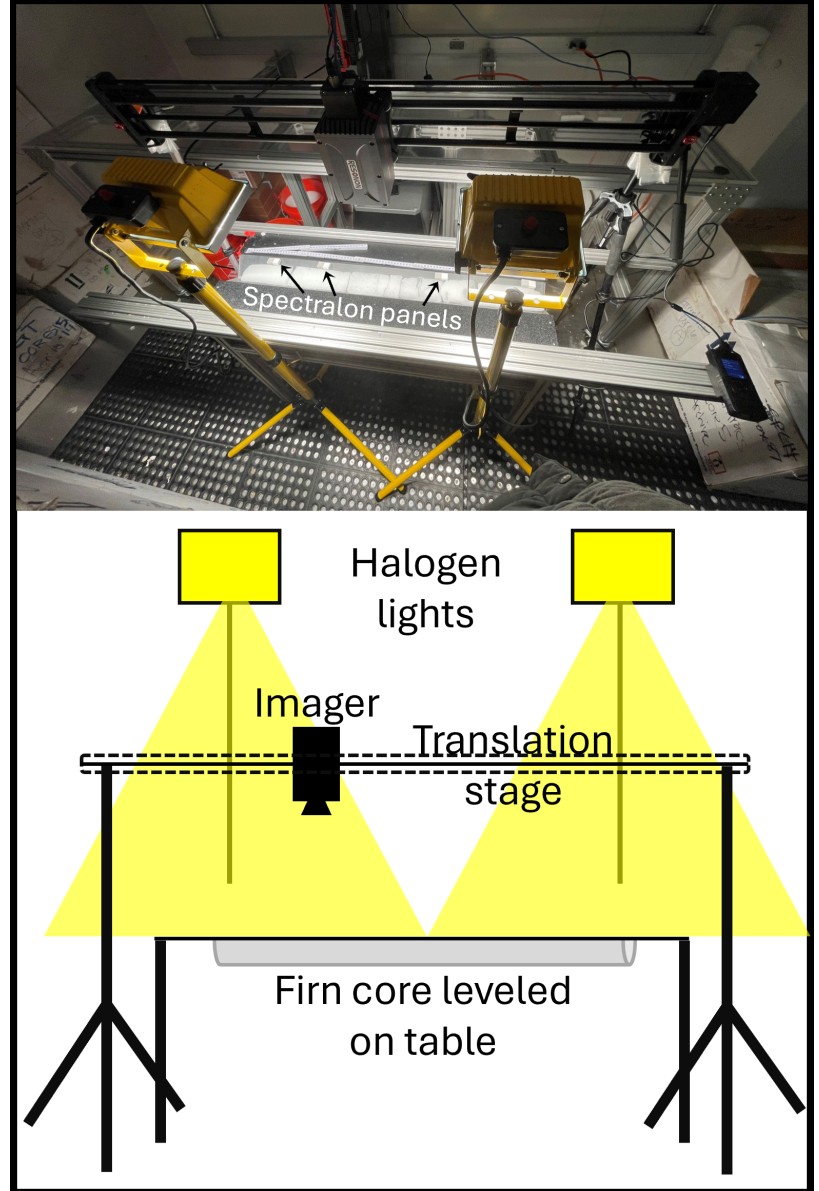

**Figure 2.** Top: Photograph showing the laboratory setup. Bottom: diagram of the laboratory setup. The imager was mounted on the translation scanning stage that moves linearly along the firn core segment. The core is illuminated by halogen lamps to provide broadband radiation. Spectralon panels were placed along the side of the firn core to convert measured radiance to reflectance.

because the core scans were collected over a period of weeks, we needed to refocus the imager at the beginning of each laboratory session. Like Donahue et al. (2021), the frame rate of the imager was set to 124 Hz and we set the integration time 165 to 8.05 ms.

The spectral data collected by the imager do not have physical units, and are recorded as digital numbers. The imager will automatically remove the dark current from the scans after a dark correction is performed. We performed a dark correction at the beginning of each scanning session by recording multiple frames with the lens cap on. After the scans were completed, the data were post-processed using the Resonon Spectronon proprietary software, where the raw data were converted to radiance using the calibration file generated by Resonon for the specific imager and objective lens, which removed the instrument-sensor-response function from the data. Finally, to convert the measured spectra from radiance to reflectance, Spectralon white reflectance panels were placed along the firn core and were captured in each scan (Figure 2).

Overall, imaging a single core in the cold laboratory took between 2–3 hours. Setting the focus of the NIR-HSI took between 20–30 minutes and required repeated scans to test the minor adjustments of the objective lens. Once the imager had been focused, unpacking and repackaging each firn core segment before and after the scan was the rate limiting step ($\sim$10 min). The scanning process itself took $\sim$10 seconds, and each core consisted of approximately 10–15 segments. Processing the images in the Spectranon Software (applying the dark correction, converting raw data to radiance, transforming radiance to reflectance) required $<$ 5 minutes.

In addition to scanning all half-round core sections to construct full-length core grain size maps and ice layer distributions, we conducted two experiments to test the sensitivity of the imager to the orientation of firn core segments and various levels of objective lens focus. We wished to determine if this scanning technique could accurately retrieve grain sizes on firn cores that had not previously been cut into half-rounds, which provides a flat face to illuminate and scan. We scanned one firn core segment with the NIR-HSI with the flat face exposed to the camera, and without changing the setup or focus, we flipped the core segment so that the curved face was scanned by the imager. We then compared the grain size profiles retrieved from our grain size inversions. Similarly, we scanned a single core segment multiple times by slightly changing the focus of the objective lens. Because the focus of the lens likely changed slightly from day to day, we wanted to ensure the grain size profiles would be similar. We compared grain size profiles from a "focused" scan and an "unfocused" scan.

## 2.3 Determining effective grain radii from measured reflectance spectra

Near-infrared snow reflectance is highly sensitive to snow grain size (Nolin and Dozier, 1993). Remote sensing studies have mapped effective snow grain radius at the landscape scale by calculating the size of ice absorption features in the reflectance spectra and relating them to the radius of an optically equivalent sphere (Nolin and Dozier, 2000). We employed the Nolin-Dozier technique (Nolin and Dozier, 2000) for determining effective grain radius, $r_e$, by using the scaled band area. The scaled band area, $A_b$, is the area between the measured reflectance, $R_m$, and the continuum reflectance, $R_c$, integrated over the ice absorption feature centered at 1030 nm and scaled by the continuum reflectance:

$$A_b = \int\limits_{962 \text{ nm}}^{1092 \text{ nm}} \frac{R_c - R_m}{R_c} d\lambda \tag{1}$$

$R_c$ represents the reflectance spectrum as if the ice absorption feature at 1030 nm were not present, and is defined as the slope between the shoulders of the ice absorption feature.

We related $A_b$ from each pixel in our images to theoretical values of $A_b$ generated from modeled snow grain reflectance spectra using the Snow, Ice, and Aerosol Radiative Transfer Model (SNICAR; Flanner et al., 2007). SNICAR requires illumination angle, $r_e$, snow layer thickness, snow density, and concentrations of light absorbing particles as inputs to simulate radiative transfer through a snowpack comprised of uniform ice spheres. We simulated a single optically thick firn layer with a constant density (600 kg m$^{-3}$), illumination angle (0°), and impurity concentration (0 ppb) (Donahue et al., 2021). Snow density negligibly affects snow reflectance (Bohren and Beschta, 1979), and mm-to-cm penetration of NIR wavelengths justifies a single model layer. While an impurity concentration of 0 ppb is not realistic for natural settings, it is an acceptable simplification for this forward modeling because light absorbing particles lower reflectance primarily in the visible wavelengths and this impact does not extend into the portion of the NIR spectrum used to retrieve grain size (Bohn et al., 2021).

We varied $r_e$ in each simulation to generate a lookup table with a theoretical $A_b$ assigned to each value of $r_e$ ranging from 0.05 to 10 mm. We retrieved a $r_e$ value for each pixel in our images by querying the pixel's measured $A_b$ value using a piecewise cubic hermite interpolating polynomial on our lookup table in MATLAB. Because of the high image resolution and large number of pixels, the inversion to retrieve grain radii lasted 5–10 minutes for each core. We directly report the $r_e$ values retrieved through this forward modeling approach, but reiterate that this is an optical property; because of the field of view, and mm-to-cm penetration of NIR light, the resultant pixel reflectance can represent light interactions with multiple grains.

## 2.4 Analyzing firn core images

### 2.4.1 Image cropping

Each 10 m core consisted of 10 – 16 segments ≤ 1 m long, which required a separate image for each section to document a full core. After the images were collected, they were then cropped in the Spectronon software to remove the outer 1-2 cm from the side edges and the top/bottom ~1 cm. Cropping the ~8 cm-wide firn cores resulted in images and grain size maps that are ~5 cm wide. We cropped the images to remove spurious grain size gradients along core edges that resulted either from slight illumination variations or the loss of some light to transmission as the thinner firn core edges approach the thickness of the light penetration depth. Some bands of anomalously small retrieved grain sizes appear in the maps that have not been removed from cropping, since we attempted to strike a delicate balance between removing lighting artifacts and not cutting a significant portion of the firn core from the images. After the grain radius inversion, we rotated the grain size maps from horizontal to vertical and vertically stacked each segment to recreate the full 10 m core. Once stacked the full image of the core was ≤10 cm shorter than the real core, an artifact of the edge cropping of individual images. We assigned a depth array equal to the number of rows of pixels in the image with the start and end depth equal to the visually-identified top and bottom depths of the core. In effect, by equating the depths this slightly stretched the core image, and the depth uncertainty of features in maps created from the images ~2–5 cm. We note that this is approximately the same as the uncertainty in the depths of visually-identified features, since it is often difficult to measure exactly where ice layers begin/end, and the top and bottom of core segments are often jagged or deteriorated, which makes it challenging to accurately set the length of each core segment. These uncertainties can result in slight depth discrepancies between visually-identified and hyperspectrally-retrieved firn core stratigraphy.

### 2.4.2 Developing a grain radius threshold to identify infiltration ice

Pixels containing infiltration ice from refrozen meltwater are immediately apparent in the raw grain size retrievals as anomalously large radii compared to the surrounding firn grains. We developed an effective grain size threshold to classify infiltration ice features in the cores in order to both remove them from the maps to prevent them from biasing average grain sizes, and to create explicit maps of ice layer stratigraphy.

We first calculated the total percentage of infiltration identified visually in each core by summing the fractional extent of each ice layer in the core (expressed as a total height of ice) and dividing by the length of the full core (Eq. 2):

$$\text{VI} = 100 \times \frac{\sum (H_n \times W_n)}{L} \tag{2}$$

where *VI* is the percentage of infiltration ice identified through visual inspection, $H_n$ is the thickness of ice layer *n*, $W_n$ is the width of ice layer *n* expressed as a fraction of the core width, and *L* is the total length of the core.

We compared *VI* to the percentage of infiltration ice in the hyperspectral images. To quantify the hyperspectral infiltration ice, *HI*, we classified pixels as either infiltration ice or firn based on a threshold grain radius (Eq 3):

$$\text{HI} = 100 \times \frac{P_i}{P_t} \tag{3}$$

where *HI* is the percentage of infiltration ice in the hyperspectral images, $P_i$ is the number of pixels classified as ice, and $P_t$ is the total number of pixels in the full core image.

We leveraged our dataset of visual ice layer stratigraphy to determine the threshold grain radius characterizing infiltration ice. We iterated over a range of possible threshold grain radii ranging from 0.7 – 1.3 mm with a step size of 0.1 mm, and categorized any pixel with a retrieved radius greater than the threshold as infiltration ice and calculated *HI* for each core. We regressed *HI* versus *VI* for all cores and calculated the root mean square error (RMSE) during each iteration. Upon finding the threshold radius that minimized the RMSE of the regression (1 mm), we cycled over threshold radii between 0.95 mm and 1.05 mm, increasing the threshold by 0.01 mm for each iteration. Through this process, we minimized the RMSE between the hyperspectral infiltration ice content and the visual ice content and determined a threshold radius with a precision of 0.01 mm. An infiltration ice threshold radius of 1.04 mm minimized the RMSE (1.03%) (Figure 3).

We also set a lower-bound radius threshold of 0.15 mm to remove anomalously small grain sizes retrieved along breaks in core segments by by reconciling grain radii maps from the NIR-HSI and visual stratigraphy and identifying regions where grain sizes either sharply decreased at breaks in the core segments.

### 2.4.3 Comparing traditional and effective grain sizes

We compared our effective grain size retrievals to the traditional grain size measurements from Cores 1–7 generated by McDowell et al. (2023). To compare grain sizes, we averaged grain radii over the same depth bands over which McDowell et al. (2023) reported average grain diameters. We explored a simplified firn grain geometry to explain discrepancies between the traditional and effective grain sizes in each core.

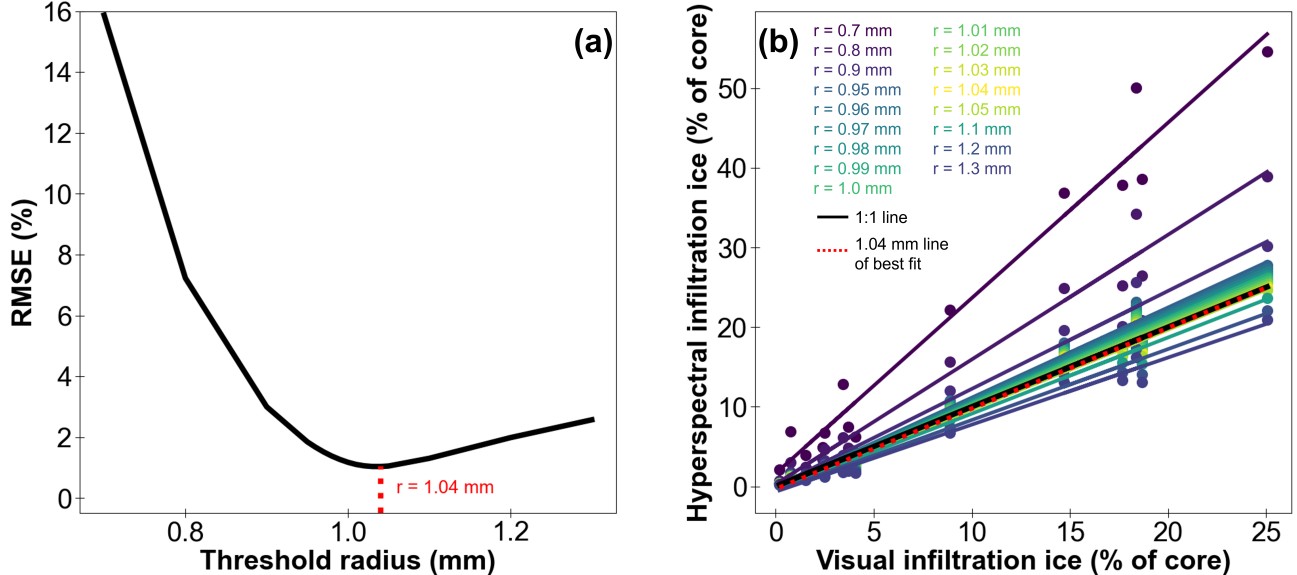

**Figure 3.** The iteration process to determine the threshold grain radius used to classify infiltration ice. (a) A threshold radius of 1.04 mm minimized the RMSE of the regressed hyperspectral infiltration ice versus the visually-detected infiltration ice. (b) Regressions of hyperspectral infiltration ice versus visual infiltration ice for each threshold radius in the iteration are shaded based on the radius used to threshold out infiltration ice. Black line denotes the 1:1 line, and the regression using the 1.04 mm threshold is shown with the red dotted line. Each point represents the infiltration ice content for different cores.

### 2.4.4 Characterizing impacts of the 2012 melt event on firn structure

Using the depth–age scales developed by Graeter et al. (2018) and Lewis et al. (2019), we examined structural changes to the firn column created during the extreme melt event of 2012 to demonstrate the utility of these detailed firn stratigraphy maps. We selected a time window that spanned firn ages where the firn would have been most directly impacted by surface melting in the summer of 2012. Our temporal window comprising the 2012 melt layer spanned from 1 January 2011 to 1 September 2012 (hereafter 2012 melt layer). We chose these dates because most of the summer melting would have concluded by September,

and the lower bound of 1 January 2011 ensured that the entire previous year of firn would be included. While meltwater can percolate deeper than the firn from the previous year (e.g., Humphrey et al., 2012; Charalampidis et al., 2016), near-surface firn will experience the strongest effects from wetting fronts and the heterogeneous piping events occurring before the wetting front arrival (Humphrey et al., 2012). We examined how grain size and infiltration ice content differed between firn within and outside of the 2012 melt layer. We determined infiltration ice content by calculating the percentage of pixels with a grain radius

> 1.04 mm. Additionally, we examined the deep structural changes caused by the 2012 melt event by comparing ice content

and grain sizes in firn deposited before the end of the 2012 melt layer (1 September 2012) to sections deposited after the 2012 melting ceased.

## 3 Results and discussion

### 3.1 Grain size retrieval sensitivity to objective lens focus and firn core curvature

We ensured that grain size retrievals were not affected by slight changes in the focus of the objective lens. Focusing NIR-HSI is both the most time-consuming step of the imaging process and introduces some level of subjectivity as the user decides when an image appears most focused. It is therefore difficult to achieve the same level of focus in each scan. We compared grain size retrievals from two images of the same core segment: one image where the core appeared to be in-focus, and one where the image of the core was blurred. Mean grain sizes retrieved in the two images are nearly identical (Figure 4a). Differences in grain

sizes are randomly distributed throughout the maps, except for where prominent ice layers exist. The grain size differences near these ice features may be caused by a slight blurring of the ice layer edges or a very small image misalignment, which may have been introduced when cropping the different images. However, the randomness of the grain size biases in undisturbed firn is similar to the spatial pattern of random sensor noise (Donahue et al., 2021). The random noise in grain size differences average out to produce nearly identical mean grain size profiles, and should not skew grain size distributions generated from

either map. The largest difference in mean grain size profiles is 0.008 mm, and the average difference across the entire core segment is 0.002 mm, both within the range expected from random sensor noise (Donahue et al., 2021). While the focus level may vary slightly between firn core images, it does not effect the interpretation of the grain size retrievals.

Because many firn cores are not immediately cut into half-round sections when returned from the field, we tested how sensitive grain size retrievals are to the curvature of the firn core surface. We aimed to evaluate whether significant biases were

introduced by scanning cores that had curved surfaces to determine whether firn cores need to be cut into half-round sections for this imaging methodology to provide reliable data.

Grain size maps look similar from images of firn core segments with flat and curved surfaces (Figure 4b). However, when subtracting the map of grain radii retrieved from the curved core image from the map of grain sizes from the flat core surface, a clear cross-width gradient in grain size differences emerges. The grain size gradient results from illumination variation across

the curved surface caused by the lighting source being slightly off-nadir to prevent a shadow from the NIR-HSI. If the light source were nadir, highest reflectance (and thus lowest grain size) should be found in the middle of the core. The slightly off-nadir illumination introduces a grain size bias from left to right as one side of the core is more illuminated than the other. Mean grain size profiles show relatively good agreement (the average grain size difference across the entire core segment is 0.014 mm), the apparent consistency in mean grain size profiles results from an averaging of the positive and negative biases.

We attribute larger differences in the mean grain size profiles (0.15 mm) at 6.7 m depth to an elevated presence of infiltration ice in the bottom of the half-round core section, which faces the imager when flipped upside down to scan the curved surface. This result demonstrates the lateral variability of meltwater flow and refreezing, even on the centimeter scale within firn cores.

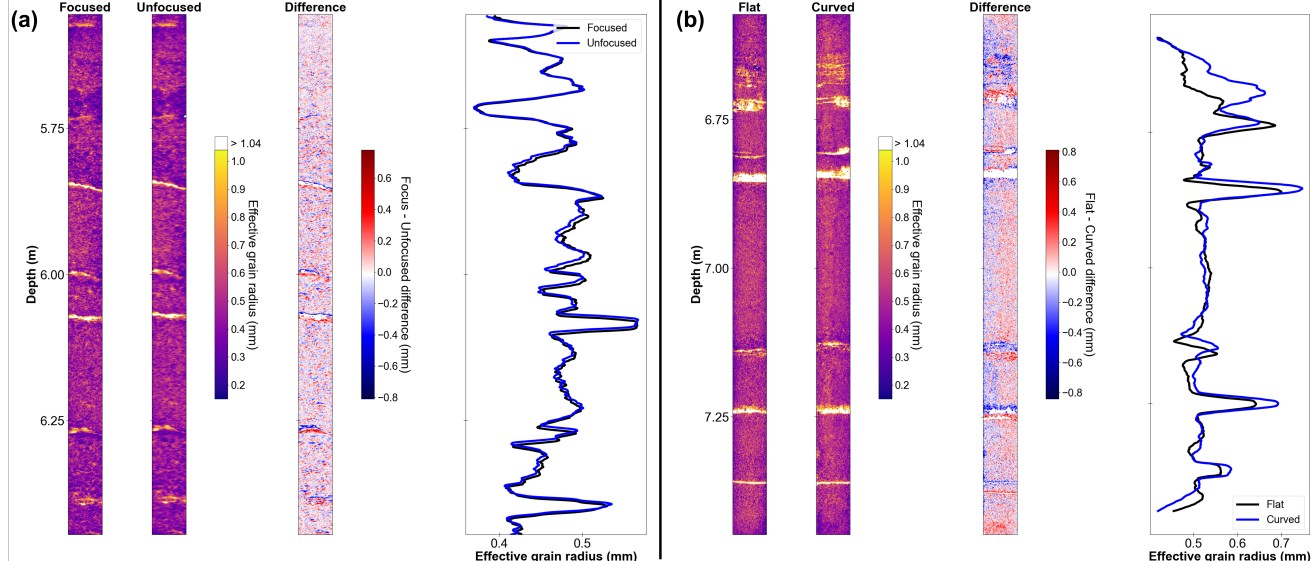

**Figure 4.** Grain size retrieval sensitivity to firn core surface curvature and objective lens focus. (a) Grain size maps, the map of grain size differences, and mean grain size profiles for firn core segments scanned with the NIR-HSI objective lens in focus and out of focus. In the difference plot, red colors show pixels where the retrieved grain size is higher in the flat core image, while blue colors represent where the grain size retrieved from the flat firn core surface is lower than from the image of the curved core segment. (b) Grain size maps, the map of grain size differences, and mean grain size profiles for firn core segments scanned with the flat half-round surface facing the NIR-HSI and the curved bottom surface facing the imager. The shading of the difference plot is the same as in panel (a). All pixels with an effective grain radius > 1.04 mm are masked as infiltration ice.

Scanning firn cores that have not been cut into half-round sections will produce similar overall grain size profiles, and may be useful for infiltration ice feature identification; however, detailed analysis of firn grain sections should only be conducted
using maps produced from images of flat firn core surfaces. The magnitude of grain size biases are relatively high (∼0.5 mm), and unevenly distributed across the core, and would affect grain size distributions. This may skew interpretations of grain size distributions; for example, preferential flow paths that have caused grain coarsening can produce bimodal grain size distributions (Avanzi et al., 2017). Grain size distributions where grain size transitions create capillary barriers are also bimodal (Donahue et al., 2021), which may artificially appear in distributions generated from maps of grain size over curved firn core
surfaces when preferential flow pathways or capillary barriers are absent. While a nadir light source, e.g., lamps mounted directly around the imager's lens similar to the setup in Donahue et al. (2022), might remedy the cross-core gradient in retrieved grain size, imaging a curved core with the NIR-HSI will still present challenges. The curvature of a firn sample changes the angular distribution of reflected radiation. Along the core edges, the grazing angle of the illumination source becomes large, which will reduce measured reflectance from forward scattering. Therefore, we recommend any detailed analyses of firn
structure using the NIR-HSI be conducted on half-round firn cores.

## 3.2 High-resolution grain size maps

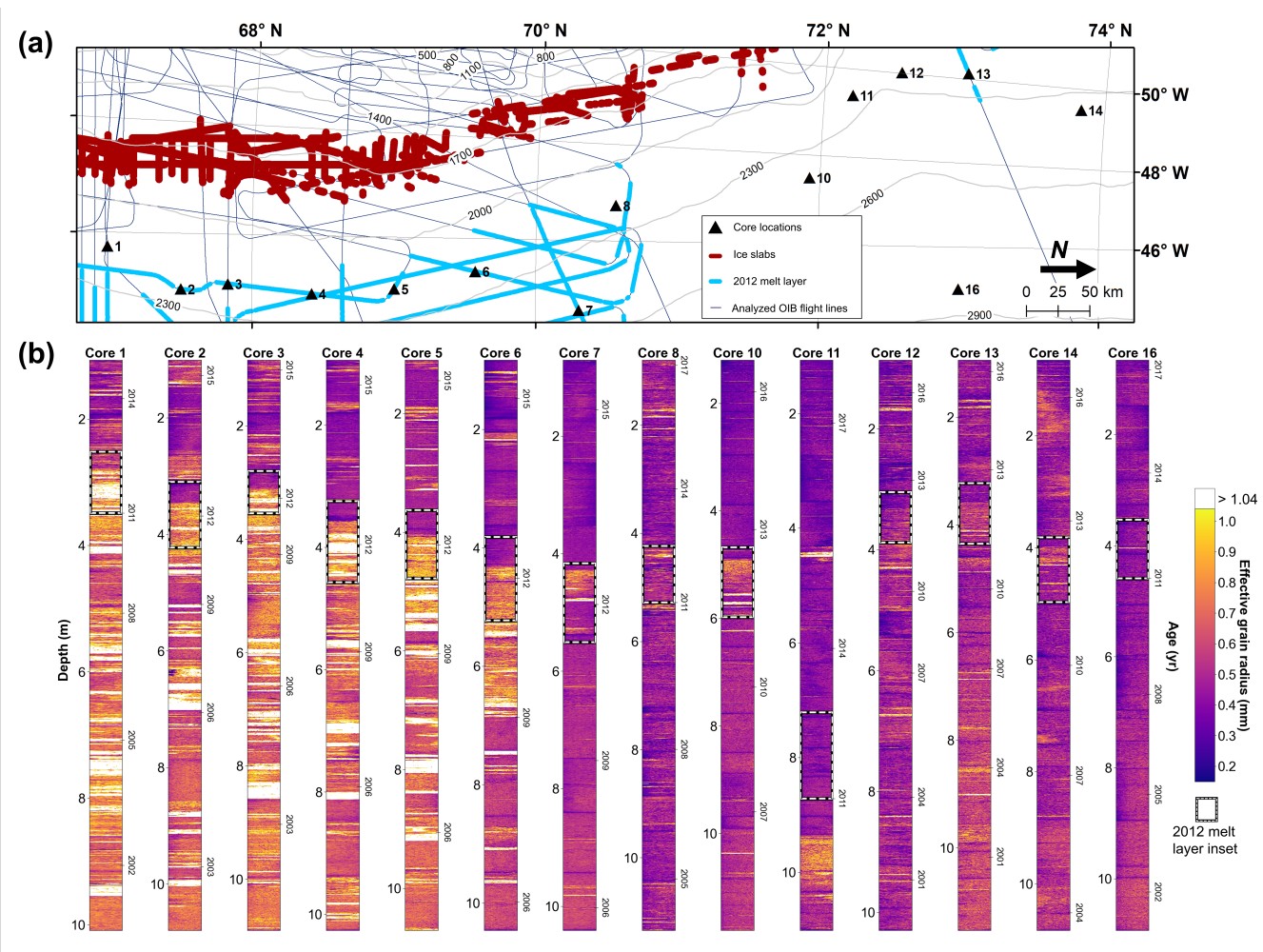

**Figure 5.** Firn core stratigraphy maps. (a) Inset map from Figure 1 with firn core locations labeled in black, impermeable ice slab extents in red (MacFerrin et al., 2019), and the 2012 melt layer detections in navy blue (Culberg et al., 2021). Operation IceBridge (OIB) flightlines analyzed by Culberg et al. (2021) are displayed as thin blue lines. (b) Firn core stratigraphy shaded by effective grain radius ($r_e$). Pixels with an effective grain radius >1.04 mm are classified as infiltration ice and masked. The black and white dashed extent indicators denote firn deposited between January 2011 and January 2013, which should have been affected by the extreme melt event of 2012, that are shown in Figure 8.

After demonstrating that images from the NIR-HSI produce robust estimates of grain size over half-round firn core segments regardless of slight changes in focus, we mapped firn core grain size and ice layer stratigraphy in the 14 cores along the GreenTrACS transect (Figure 5). The pixel size in each scan is approximately 0.4 x 0.4 mm, which can capture distinct

differences in physical characteristics between adjacent layers caused by intermittent snow deposition on the ice sheet surface and subsequent metamorphism (Benson, 1962; Colbeck, 1982; Alley, 1988). Monograin ice crusts remnant from surface glazes during previous accumulation hiatuses and wind packed layers can also be preserved within the firn column and cause a local reduction in vertical permeability (e.g., Courville et al., 2007). The NIR-HSI is able to detect these monograin crusts (Figure B1), which indicates that these maps can resolve the fine-scale heterogeneity in grain scale properties required to accurately simulate gas, vapor, and fluid flow through the firn. While the maps can be useful to initialize models, they also can highlight key hydrologic processes. In many of the core maps, preferential flow pathways are apparent as tortuous sections of elevated grain size, caused by wet grain metamorphism as water percolated through some sections of the core with the surrounding firn remaining dry (Figure B1). These preferential flow paths are critical for the development of uniform wetting fronts and also deep ($> 10$ m) meltwater percolation in firn (Humphrey et al., 2012).

Within these firn core maps, a latitudinal gradient in grain size and the number of ice layers appears, as southernmost cores have large amounts of infiltration ice and larger grain sizes (Figure 5). This effect is largely a result of the temperature gradient which increases the surface meltwater supply in southern cores (McDowell et al., 2023).

### 3.3 Traditional and effective grain size comparisons

We examined differences between traditional grain size measurements from McDowell et al. (2023) and effective grain radii from this study to understand the relationship between these two types of grain size definitions in firn. Traditional grain size measurements are often taken in the field, and these types of measurements have validated grain growth parameterizations (Lehning et al., 2002) in a physics-based land-surface snow model, SNOWPACK, that has been applied extensively to simulate the evolution of the firn layer on ice sheets and ice shelves (e.g., Groot Zwaaftink et al., 2013; Steger et al., 2017; Dunmire et al., 2021; Keenan et al., 2021; Banwell et al., 2023). Furthermore, SNOWPACK also evolves effective grain size based on the description by Vionnet et al. (2012), but this parameterization is still dependent on empirical relationships with traditional seasonal snow grain size. Our effective grain size dataset provides a valuable opportunity to further investigate the discrepancies and the empirical relationship between traditional and effective grain size measurements of firn.

Across the 7 cores that have traditional grain size measurements, our effective grain size profiles show synchronous variations with depth (Figure 6a). While grain size profiles appear similar, the grain radii retrieved from the NIR-HSI are consistently larger than traditional measurements. Average effective grain sizes in each core ranged from 1.46 to 1.61 times larger than traditional grain radii, with the mean ratio of effective-to-traditional grain size across all cores being 1.51 (Figure 6b).

Effective and traditional grain sizes are not expected to be the same since they are based on different grain properties. Effective grain size retrievals based on NIR reflectance spectra rely on measurements of ice absorption and are ultimately a measurement of optical path length. Alternatively, traditional observations measure the cross-sectional extent of the grain. Previous limited studies report effective snow grain sizes to be $\sim 2 - 20$ times smaller than traditional grain radii (e.g., Painter et al., 2007; Langlois et al., 2010; Leppänen et al., 2015). However, we find that effective grain sizes are consistently *larger* than traditional grain size, while the magnitude of the effective-to-traditional grain radius differences are smaller than in snow. This suggests a different relationship between the two measurements in snow and firn.

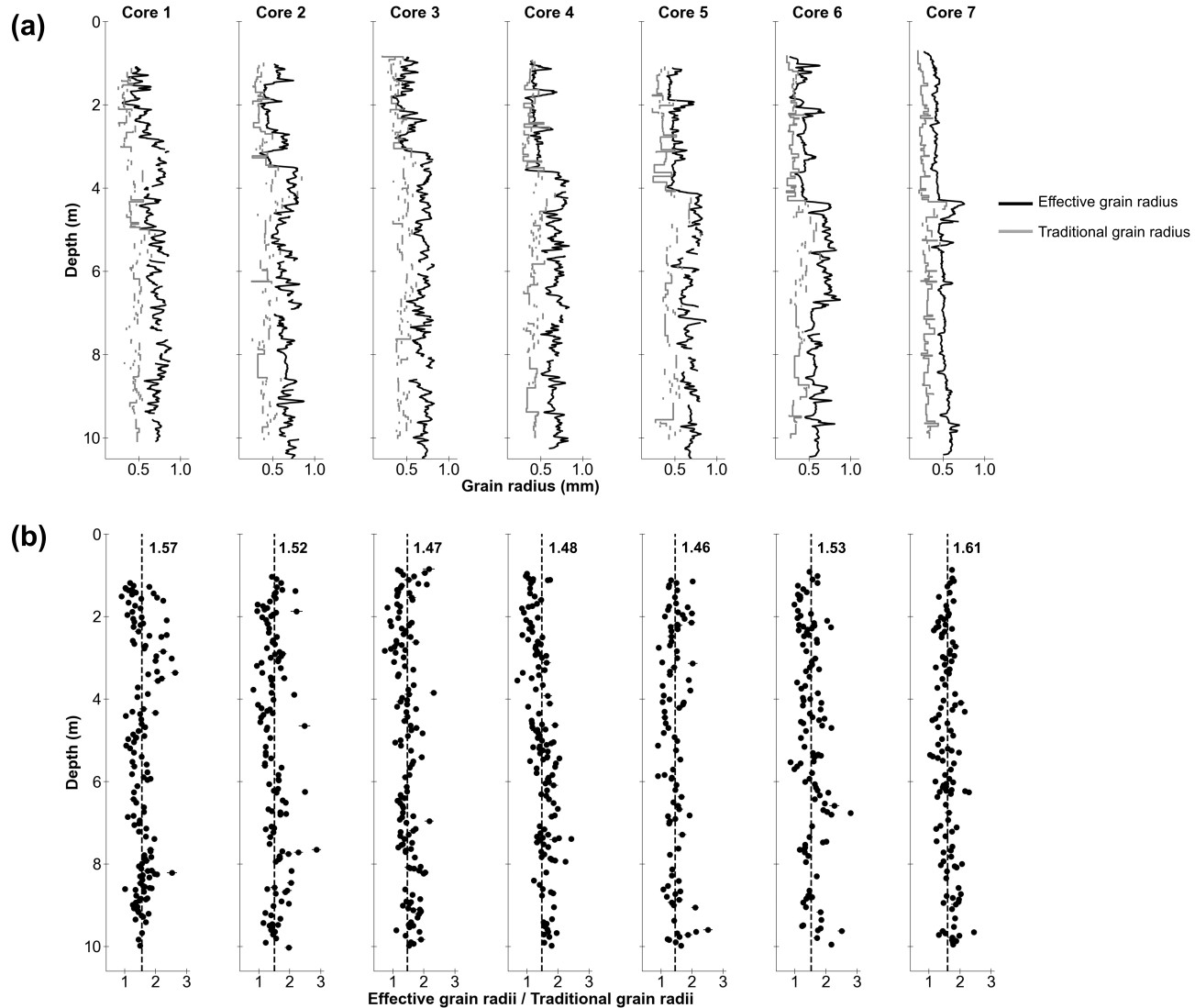

**Figure 6.** Effective vs. traditional grain sizes. (a) Grain size profiles from digital grain diameter measurements from McDowell et al. (2023) (grey) and from the NIR-HSI (black). (b) Ratios of effective grain sizes to traditional grain sizes. The mean ratios for each core are reported and represented by the dashed black lines.

The high bias in effective grain size retrievals compared to traditional observations of firn grains is likely caused by the

355 methodological differences in measurement techniques compounded by the shapes of firn grains. While firn grains are typically spherical or spheroidal (Alley, 1997; Meussen et al., 1999), similar to the shape assumed to generate our grain size lookup table, sintering processes further reduce total surface area as firn grains become bonded together. In relatively low density firn ($\sim$350 – 550 kg m$^{-3}$), bonds between grains will form primarily from grain-to-grain vapor diffusion until the radii of curvature become

large enough that surface diffusion becomes an important mechanism driving neck growth (Maeno and Ebinuma, 1983). The lower specific surface area of bonded grains, where the space between the grains has become ice-filled compared to two individual grains with an air-filled grain boundary, will cause greater NIR absorption and thus the retrieval of a larger sphere with the same surface-to-volume ratio (Wiscombe and Warren, 1980). The discrepancies between effective and traditional grain sizes can be further increased when the firn cores are cut into half-round sections. It is highly unlikely that all firn grains are cut exactly through the center. Therefore, a traditional grain size measurement will calculate the diameter of a smaller cross-section than if the spherical grains were cut through the middle. Because NIR wavelengths penetrate to a depth of a few grain diameters, artifacts introduced by firn core cutting will not influence the absorption and resulting grain size retrieval. Therefore, in firn, we largely attribute the >1 ratio of effective-to-traditional grain size to both increasing the retrieved effective grain radius by firn grain bonding that decreases specific surface area of the firn medium, and decreasing the traditional grain measurement by firn core cutting that exposes cross sectional areas smaller than perfect hemispheres. Our reported biases still suggest a distinctly different relationship than seen in seasonal snow grain size comparisons. In seasonal snowpacks, large temperature gradients promote the prevalence of kinetic growth forms with high surface area-to-volume ratios. The thicknesses of these grains can be 50 times smaller than their surface extents and these forms may also be hollow (e.g., Taillandier et al., 2007). The effective grain diameter of needles and plates are similar to their thicknesses (Mätzler, 1997), while a traditional measurement of their maximum extent would be much larger, which likely results in smaller effective grain sizes compared to traditional measurements.

While we largely attribute the discrepancies between effective and traditional firn grain size to the fact that NIR reflectance is governed by ice absorption and traditional grain size is controlled by observable cross-sectional area, we notice that a simple geometric correction relating the grain shape proposed in the grain scale model of meltwater movement through firn by Humphrey et al. (2021) can explain a majority of the effective-to-traditional grain size differences in our dataset. Because the truncated octahedron grain shape proposed by Humphrey et al. (2021) assumes a highly-idealized firn grain, we dedicate a discussion developing a geometric-based correction factor relating effective and traditional grain size in Appendix A.

### 3.4 Comparing mapped ice layer distributions to visual ice layer stratigraphy

We used our firn scans to generate maps of ice layer stratigraphy, which we compared to ice layer distributions generated by visually inspecting firn cores on a light table to determine if the two methods generate similar data. Challenges with developing ice layer distributions through visual inspection arise if ice layers are thin and difficult to see, and it can be complicated to discern the full extent of ice layers because surrounding firn grains limit the ability to see into the firn core. Infiltration ice in the grain size maps are readily apparent as pixels with grain radii larger than the fine- to medium-grained surrounding firn. We could easily and quickly binarize the scans into firn or ice using a threshold grain radius and produce explicit ice layer maps (Figure B2). Given the ease of mapping ice layers using the NIR-HSI, we wanted to characterize any discrepancies between this method and visual inspection.

We find that these ice layer maps match the ice layer distributions that we documented by visually inspecting cores on the light table (Figure 7). There are slight vertical offsets between infiltration ice identified in hyperspectral images and on the

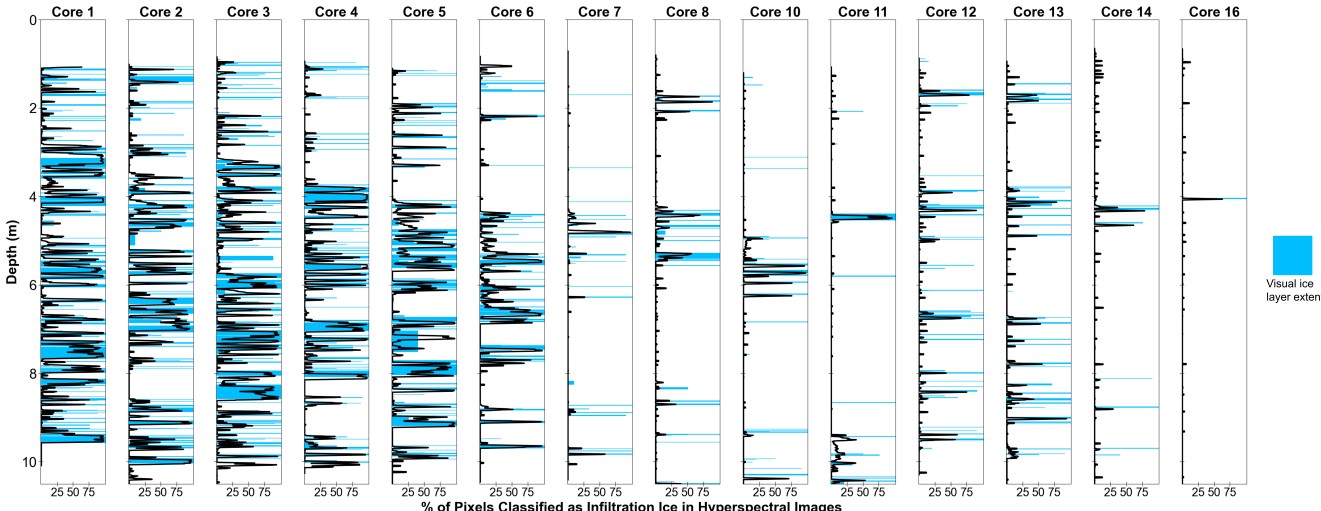

**Figure 7.** Comparison of infiltration ice mapped by the NIR-HSI and identified by visual inspection. Profiles from the NIR-HSI are represented as the line-by-line percentage of pixels classified as infiltration ice (black line corresponding to the x-axis). The horizontal and vertical extent of ice layers identified on a light table is shown in blue.

light table. These slight discrepancies arise from small (cm) uncertainty in the depths of these features introduced during both the cropping of hyperspectral images and logging core stratigraphy on a light table as described in Section 2.4.1. Additionally,
spurious increases in infiltration ice content mapped by the NIR-HSI in Figure 7 appear where breaks in the firn core segments occur. While these excursions introduce noise to the infiltration ice profiles, they are not large enough to impact calculations of total infiltration ice in each core. The binarized ice layer maps reproduce the amount of infiltration ice in each that we identified visually ($R^2$ = 0.985, p $<$ 0.001; Figure B3). This is a promising result, which demonstrates that the NIR-HSI produces ice layer distributions very similar to what would be described through visual inspection, while also providing valuable grain size
data. However, this technique requires choosing the correct threshold radius to classify ice layers. These firn cores consisted of fine-to-medium sized firn grains according to the classification scheme by Benson (1962), which allowed for sections of infiltration ice to immediately stand out in raw grain size retrievals. Thresholding grain size maps in coarse-grained firn may prove more difficult. Infiltration ice content identified in the firn core images from the NIR-HSI is slightly higher than what is visually noticed in firn cores where summer air temperatures are highest (Figure B3). This discrepancy could either be caused
by the difficulty to accurately visually quantify many small ice lenses, pipes, or ice layers in these cores with a high supply of surface meltwater, or our threshold value for ice layers might categorize some regions of wetted firn that have large grain clusters as infiltration ice, but wetted firn would not be classified as an ice layer on a light table. For example, it is difficult to determine the infiltration ice content of a preferential flow path, such as in Figure B1, on a light table because it is difficult to see the entire extent of the flow path; however, in our maps, many pixels within the flow path are categorized as infiltration ice.

## 3.5 Impacts of the 2012 extreme melt event on firn structure

Our previous analyses demonstrate that the NIR-HSI produces reasonable maps of firn core stratigraphy based on the agreement between effective firn grain sizes from this study with traditional grain size measurements and similar distributions of infiltration ice features as developed through visual inspection. We illustrate the utility of these high-resolution firn stratigraphy maps by analyzing structural changes to the firn column induced by the 2012 extreme melt event.

In almost every core, the 2012 melt layer consisted of a sudden increase in grain size, an increased presence of ice layers, or both (Figures 5, 8). With the exception of Cores 6, 11, and 12, all of the firn cores contained elevated levels of infiltration ice within the melt layer compared to the rest of the core (Figure 8c). Grain sizes within the 2012 melt layer were larger than the surrounding firn in Cores 2 and 4 – 10 ($p < 0.001$; Table B1). Elevated grain sizes in firn can indicate previous firn wetting (McDowell et al., 2023), as firn grains grow rapidly in the presence of liquid water (Brun, 1989) and form clusters over successive freeze/thaw cycles (Colbeck, 1982). We suggest that the Cores 1 and 3 did not exhibit significantly larger grain sizes within the melt layer, because these cores at the southern end of the transect received the most substantial meltwater inputs (McDowell et al., 2023), so wet grain growth in firn outside of the 2012 melt layer would mask the signal from 2012 melt alone.

Structural changes to the firn column from surface melting in the summer of 2012 were detected in radar soundings from ~9000 km of flight survey lines across all sectors of the Greenland Ice Sheet (Culberg et al., 2021). The lack of flight coverage or radiometric issues with the radar data prevented melt layer a detection at 5 core sites (Cores 10, 11, 12, 14, and 16) (Figure 5a). Given that surface melting occurred across 97% of the Greenland Ice Sheet during this extreme summer (Nghiem et al., 2012), evidence of meltwater percolation should be apparent within the shallow firn column across most of the ice sheet. Our stratigraphic maps provide evidence of a 2012 melt layer that is structurally-different from the surrounding firn column in Cores 10, 14, and 16. Each of these cores have elevated levels of infiltration ice within the 2012 melt layer compared to the surrounding sections of firn, and grain sizes in Core 10's melt layer are much larger than in the surrounding firn (Figures 5b, 8; Table B1). While we show evidence of a 2012 melt layer in regions that were not analyzed by Culberg et al. (2021), we also find a thick ice layer from the 2012 melt event in Core 1 that was not detected in the radar data. The detection algorithm of Culberg et al. (2021) was based on firn density contrasts that generate powerful radar reflectors. Culberg et al. (2021) speculated that the absence of the 2012 melt layer at lower elevations in the percolation zone was likely a result of increased background levels of infiltration ice in the firn column that prevented ice layers formed in 2012 from generating a sufficient density contrast. Evidence of the 2012 melt layer in these images while being undetected in the radar data appears to confirm this supposition.

Core 11 is the only core in which we find no evidence of previous firn wetting during 2012 (Figures 5b, 8); there are no statistical differences between grain sizes and the amount of infiltration ice in firn possibly affected by 2012 melt. Instead, there is a ~1 m thick section of enlarged firn grains and infiltration ice consistent with previously-wetted firn ~0.5 m below the analyzed region of interest that corresponds to firn deposited in 2009–2010. We suspect that the extreme melt season in 2010 created this structurally-different firn layer (Tedesco et al., 2011). The complete absence of any melt signal from 2012 in Core 11 is likely driven by its unique climate setting. There is a high accumulation region near Core 11 that nearly doubles

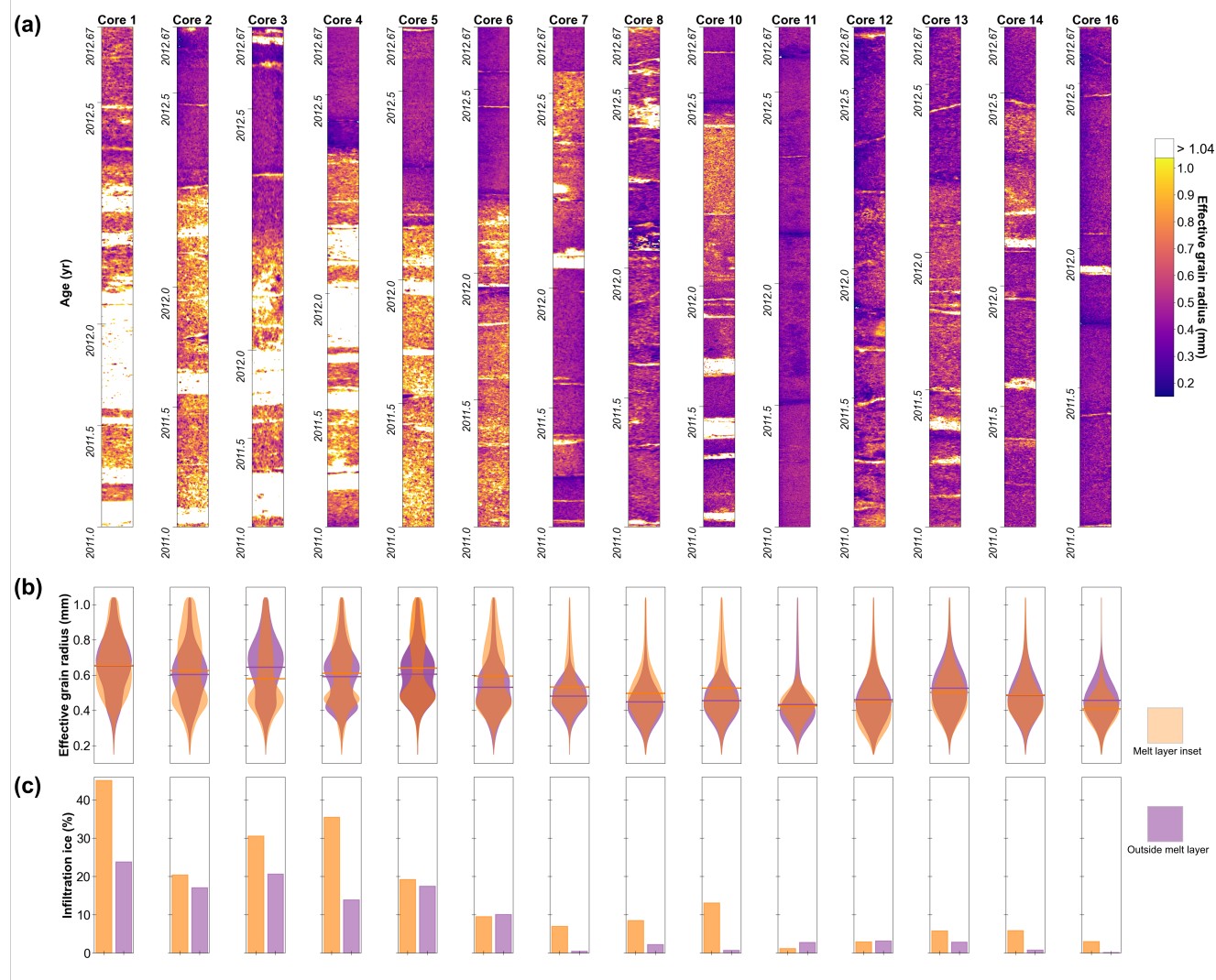

**Figure 8.** Firn structure within/outside of the 2012 melt layer. (a) Grain size and ice layer maps of firn spanning the 2012 melt layer from 1 September 2012 to 1 January 2011. Pixels with a retrieved effective grain radius >1.04 mm are classified as infiltration ice and masked. The thin bands of smaller effective grain size retrievals are artifacts caused by lighting effects at the ends of firn core segments that have not completely been removed by image cropping. (b) Violin plots showing grain size differences from firn within the melt layer (orange) and outside of the melt layer (purple). Horizontal lines represent the means of the grain size distributions. (c) Bar charts quantifying the amount of infiltration ice found within the 2012 melt layer (orange) and outside of the melt layer (purple). Table B1 contains values of the mean +/- standard deviation grain sizes and total infiltration ice content in firn within and outside of the 2012 melt layer.

the average accumulation rate at this site compared to neighboring cores (Lewis et al., 2019). In 2012, the accumulation rate at Core 11 (0.61 m w.e. yr⁻¹) was almost 3x that at Core 12 (0.25 m w.e. yr⁻¹) and also exceeded the accumulation rate at

its southern neighbor, Core 10 (0.53 m w.e. yr$^{-1}$) (Lewis, 2021). Given that Core 11 was collected in a distinctly different accumulation regime, we suspect that local melting at this location is also different from other cores, which could explain the absence of a 2012 melt layer here.

While our firn images revealed the 2012 melt layer in locations not mapped by Culberg et al. (2021), we did notice similar location-dependent characteristics of the melt layer that Culberg et al. (2021) observed. The melt layer in southern cores contains multiple ice layers scattered throughout large regions of elevated grain sizes, while infiltration ice and grain growth in northern, more interior cores is more vertically-confined (Figures 8, B2). The structural differences of the melt layer across the transect are likely related to climatic differences that influence firn temperature and cold content. In cold firn, meltwater ponding along grain size transitions between adjacent firn layers likely freeze quickly and produce narrow bands of ice layers, while in warmer firn, water ponding will have enough time to initiate preferential flow paths and subsequent grain growth will allow matrix flow to develop into deeper regions of firn (e.g., Hirashima et al., 2019). More detailed hydrological modeling will be required to confirm this hypothesis.

The 2012 melt event likely impacted firn structure in layers deeper than the previous year of firn, as refreezing preferential meltwater percolation has been documented multiple meters below the surface in subfreezing firn (Humphrey et al., 2012; Charalampidis et al., 2016). Using our firn images, we can describe the permanent and deep changes to firn structure from the 2012 melt event. There is a distinct and significant difference in grain sizes as a result of meltwater percolation. Firn grains in the top portion of the firn column deposited after 1 September 2012 are significantly smaller on average in each core than grains in older firn (Figure 9a; Table B2). While firn grains grow over time without the presence of liquid water (e.g., Gow, 1969; Linow et al., 2012), the abrupt changes in grain size suggest rapid grain growth occurring when the liquid water content is elevated (Brun, 1989). Additionally, we find older firn subjected to 2012 meltwater inputs contain a higher percentage of infiltration ice than in younger firn in every core except for Core 16 where it is equal (Figure 9b). While these changes can be explained by deeper infiltration of meltwater generated in 2012, we also note that previous melt events such as in 2010/2011 could contribute to this signal. The absence of melt features in young firn corroborates the finding of Rennermalm et al. (2022) who documented a decrease in shallow firn density and ice layers in cores from Greenland's southwestern percolation zone. These observations suggest firn can regenerate meltwater storage capacity during periods without exceptional surface melting (van den Broeke et al., 2016; Rennermalm et al., 2022).

## 4  Conclusions

We demonstrated the ability of a NIR-HSI system in a cold laboratory to reliably retrieve sub-millimeter resolution grain size data and ice layer stratigraphy from 14 firn cores collected in western Greenland's percolation zone. Inverting spectral reflectance measured by a hyperspectral imager reduces time and subjectivity in grain size estimates for firn, which are needed for remote sensing, hydrological, and paleoclimatological firn applications.

We found that this hyperspectral imaging method robustly quantifies infiltration ice content compared to visual inspection of the firn cores. Although the objective lens focus varied slightly between scans, we found that grain sizes should not be

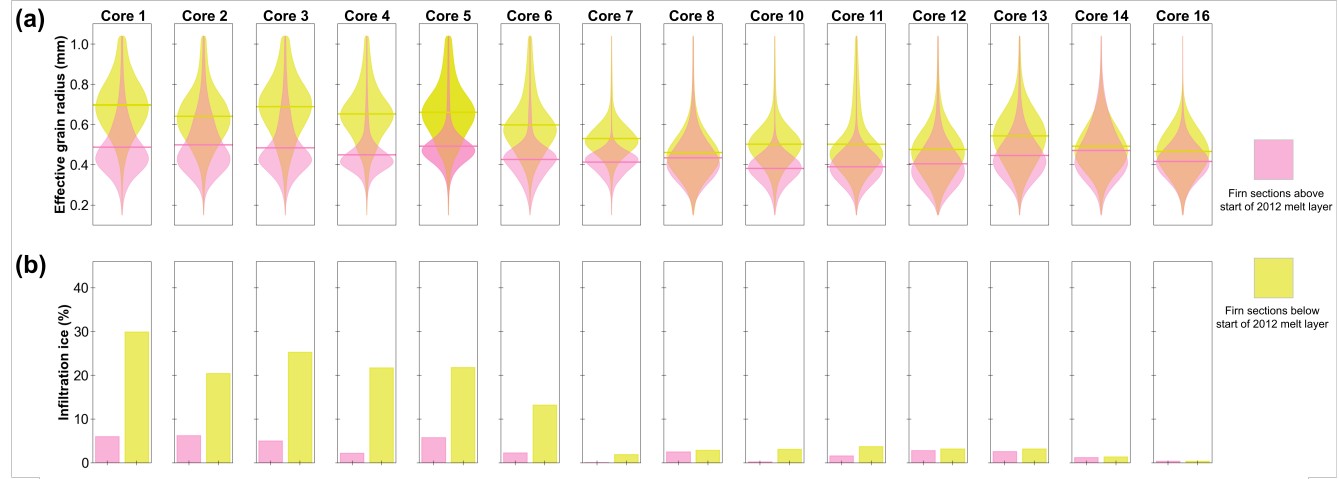

**Figure 9.** Changes to firn structure after the 2012 melt event. (a) Violin plots showing grain size differences between firn above the 2012 melt layer (i.e. deposited after 1 September 2012) (pink) and firn below the start of the melt layer (i.e. deposited after the summer of 2012) (yellow). Horizontal lines represent the mean of the grain size distributions. (b) Bar charts quantifying the amount of infiltration ice found in firn above the 2012 melt layer (pink) and below the 2012 melt layer (yellow). Table B2 contains values of the mean +/- standard deviation of grain sizes and total infiltration ice content in firn above and below the start of the 2012 melt layer.

significantly biased. However, we did show that scanning a firn core that has not been cut into a half-round section can introduce grain size gradients as a result of illumination variations across the width of the core. There is good agreement between ice layer maps and full-core percentages of infiltration ice as they are nearly identical between both methods. Compared to traditional grain size measurements collected on 7 of the same cores, the retrieved grain radii from this study are consistently larger, which is opposite of the grain size comparisons in previous studies conducted on snow. We suggest that traditional grain sizes can be converted to effective grain size by a correction factor of ~1.51. The larger effective firn grain sizes are largely attributed to th different measurement techniques; grain size based on NIR reflectance is a measurement of ice absorption while traditional grain sizes are a measurement of the grain cross-section. We do note that the treatment of an idealized firn grain as a truncated octahedra provides a geometric correction factor consistent with effective grains being 1.5x larger.

Using the maps of firn core grain size and ice layer stratigraphy, we identified the 2012 melt layer in locations where it was undetected in analysis of airborne radar surveys. Our firn dataset suggests melt layers can be identified as ice layers, abrupt changes in grain size, or both. The maps allowed us to observe and quantify how the 2012 melt layer impacted deep firn structure by elevating levels of infiltration ice and increasing grain size in firn deposited before September 2012. The maps suggest that the firn layer has experienced a decrease in melt features between 2012 and 2016/2017 when these firn cores were collected as ice sheet wide surface melting returned to the long-term average after 2012, which is consistent with other observations in western Greenland.

Detailed firn grain size datasets such as this can provide additional insight into firn densification processes, and can refine remotely-sensed mass balance interpretations, validate model simulations of firn evolution, and improve descriptions of gas diffusivity for paleoclimate reconstructions from ice cores. Given the availability and transportability of NIR-HSIs, these instruments may provide opportunities to map firn microstructure in field settings, which would prevent post-depositional effects on firn structure during transport and storage. However, difficulties in using the NIR-HSI in remote field settings may arise from creating proper illumination conditions and keeping the imager warm when not in use to prevent the optics from becoming misaligned in conditions outside of the recommended range of operating temperatures. Because firn grain size is important for many ice sheet research applications, we encourage the use of these systems in the lab or possibly in the field to constrain firn grain size across a wide variety of ice sheet settings.

*Author contributions.* IEM, KMK, SMS conceptualized the study. SMS provided the instrument and IEM conducted the lab work. CPD provided the inversion code and generated the grain size look up table. ECO, RLH, HPM collected and returned the firn cores used in this study. IEM led the manuscript writing, with input and contributions from all co-authors.

*Competing interests.* KMK is a member of The Cryosphere editorial board.

*Acknowledgements.* Collection of firn cores during GreenTrACS was supported by the National Science Foundation under Office of Polar Programs—Arctic Sciences Grants 1417678 and 1417921. Field support was provided by the Polar Field Services, and drilling equipment was loaned from the U.S. Ice Drilling Program. The authors would like to acknowledge field assistance during collection of these cores from Gabe Lewis, Karina Graeter, Tate Meehan, Thomas Overly, and Forrest McCarthy. We thank Colin Meyer for ordering lab supplies and helpful suggestions for the laboratory setup, and Wendy Calvin for lending Spectralon panels. We appreciate the correspondence with Riley Culberg to reproduce the OIB flight lines in Figure 5. The authors appreciate the assistance in the cold laboratory from Jacob Chalif, Liam Kirkpatrick, Meredith Parish, and Mikey Robinson. Thanks to Florent Domine, Nicolas Stoll and one anonymous reviewer for providing insightful and constructive comments that improved the paper.

*Data availability.* All data produced and utilized in this study are available at the NSF Arctic Data Center. Effective grain size and ice layer stratigraphy from all GreenTrACS cores in this study can be accessed at https://doi.org/10.18739/A2J38KK2T (McDowell, 2024). The traditional grain size measurements from Cores 1–7 can be found at https://doi.org/10.18739/A24B2X61Z (McDowell et al., 2022). Site mean annual air temperatures and accumulation rates, as well as the depth-age scales for each firn core can be obtained at https://doi.org/10.18739/A2X63B64H (Lewis, 2021).

## Appendix A: Explaining effective and traditional grain size differences through idealized firn grain geometry

To examine whether effective and traditional grain sizes in the firn cores can be explained by the geometric treatment of effective grains as spheres, we assume firn grains take the shape of truncated octahedra proposed by Humphrey et al. (2021) in their grain-scale model of meltwater movement through firn. While highly idealized, this shape treats firn grains as semi-rounded forms, however, they contain faceted hexagonal faces. While deep firn is isolated from large temperature-driven vapor pressure gradients, temperature data from the shallow ($< 10$ m) firn column from Humphrey et al. (2012), Charalampidis et al. (2016), and Harper et al. (2023) show that meltwater/rainwater infiltration and refreezing can create steep temperature gradients conducive to facet formation below the depth of large seasonal temperature gradients.

If the edge of a truncated octahedron has length $a$, then the length across the hexagonal face between two mid-edges is $a\sqrt{3}$ and the distance between vertices of the hexagonal face is $2a$ (Figure A1a). These two lengths are representative of the extents of firn grains measured by McDowell et al. (2023) as they digitally traced the greatest visible extents of grains. Conversely, an effective grain for a truncated octahedron can be represented as the midsphere, a sphere that is tangent to every edge of the octahedron (Figure A1b). The diameter of the midsphere of a truncated octahedron is length $3a$ (Figure A1c). The ratio of effective to traditional diameters (or radii) therefore, could range from 3/2 to $3/\sqrt{3}$, or 1.5 to $\sim$1.73.

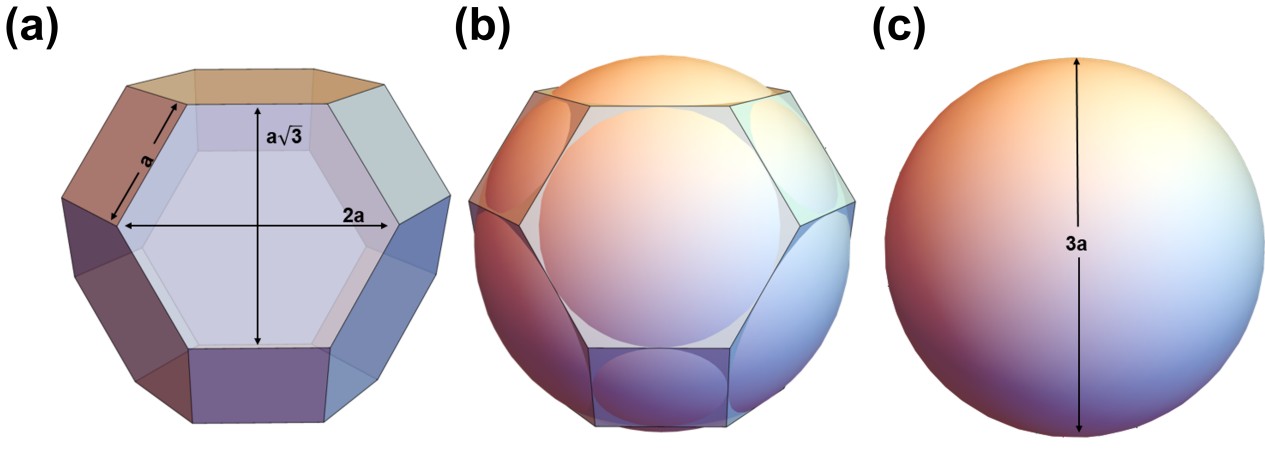

**Figure A1.** Comparison of the geometry of an idealized firn grain to an optical grain. (a) Firn grains can be represented as truncated octahedra, with edge lengths $a$. The distances between mid-edges across the hexagonal face and between vertices across the hexagonal face are $a\sqrt{3}$ and $2a$, respectively. (b) An optical grain is a sphere that fits the truncated octahedron tangent to each edge. (c) The diameter of the midsphere is $3a$.

Across all cores, the ratio of effective to traditional grain radii fell within this range in 50.5% of the core segments that had
       both traditional and effective grain size measurements (Figure A2). Additionally, effective grain radii are 1.51 times larger than
       traditional measurements on average across the entire dataset, which is very similar to the ratio of an effective grain radius
       to the length from the vertex to the center of the hexagonal face of a truncated octahedron. While the dataset contains large
       amounts of variability and the average grain size ratio in Cores 3–5 is below 1.5 (Figure A2), no core-averaged ratio is above
the geometric range.

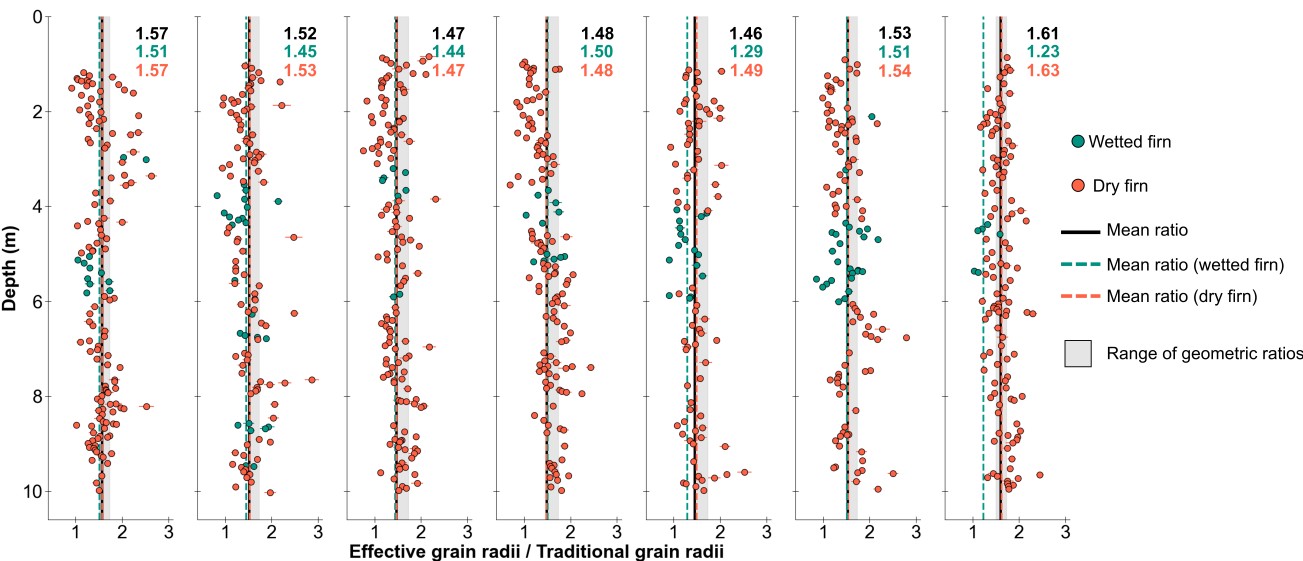

**Figure A2.** Ratios of effective grain sizes to traditional grain sizes as shown as in Figure 6b. Dry firn grain ratios are shown in orange, while
firn grains from previously-wetted regions are in teal. The dashed orange and purple lines represent the mean ratios of dry firn grains and wet
firn grains respectively. The mean ratio from the full core is shown as a black line. Each ratio for the individual cores are written and colored
corresponding to their classification. The light gray shading denotes the range of Effective/Traditional ratios expected given a hypothetical
firn grain geometry of a truncated octahedra.

       Although our proposed geometric correction factor to convert optical to traditional grain size applies for more than half of
       the dataset, the bias towards slightly lower grain size ratios in Cores 3 – 5 suggests that the firn grains may be even more
       spherical than truncated octahedra. The lower ratio than expected from the simplified geometry is likely related to effects of
       wet grain metamorphism. The effective to traditional grain size ratio is smaller in previously-wetted sections of firn than in dry
firn in all but 1 core (Figure A2). This result suggests different grain morphology between wet and dry firn, and the fact that
       wetted grains are closer to spherical confirms firn grain rounding in the presence of liquid water (e.g., Brun, 1989). In general,
       however, our results indicate that the effective grain size of shallow firn can be estimated by multiplying traditional grain size
       by a factor of ∼1.5, which can be explained by a simplified firn grain geometry.

## Appendix B: Supplementary tables and figures

**Table B1.** Grain size and ice content within 2012 melt layer compared to outside of the melt layer.

| Core | Inside 2012 melt layer | | Outside melt layer | |
| | Grain size (mm) | Infiltration ice content (%) | Grain size (mm) | Infiltration ice content (%) |
| | (Mean ± st. dev.) | | (Mean ± st. dev.) | |
| --- | --- | --- | --- | --- |
| 1 | 0.655 ± 0.167 | 45.13 | 0.654 ± 0.165 | 23.85 |
| 2 | 0.625 ± 0.200 | 20.46 | 0.602 ± 0.150 | 17.07 |
| 3 | 0.578 ± 0.189 | 30.60 | 0.644 ± 0.171 | 20.66 |
| 4 | 0.609 ± 0.187 | 35.51 | 0.591 ± 0.158 | 13.89 |
| 5 | 0.640 ± 0.198 | 19.26 | 0.605 ± 0.150 | 17.45 |
| 6 | 0.594 ± 0.181 | 9.58 | 0.531 ± 0.150 | 10.09 |
| 7 | 0.532 ± 0.142 | 7.04 | 0.480 ± 0.093 | 0.50 |
| 8 | 0.497 ± 0.143 | 8.52 | 0.447 ± 0.122 | 2.27 |
| 10 | 0.525 ± 0.167 | 13.13 | 0.454 ± 0.106 | 0.77 |
| 11 | 0.425 ± 0.079 | 1.22 | 0.433 ± 0.137 | 2.79 |
| 12 | 0.449 ± 0.150 | 2.99 | 0.460 ± 0.133 | 3.25 |
| 13 | 0.500 ± 0.141 | 5.86 | 0.525 ± 0.141 | 2.87 |
| 14 | 0.488 ± 0.137 | 5.93 | 0.484 ± 0.128 | 0.84 |
| 16 | 0.408 ± 0.095 | 3.08 | 0.457 ± 0.111 | 0.23 |

**Table B2.** Grain size and ice content in firn deposited before the summer of 2012 and after 2012.

| Core | *Firn younger than 2012 melt layer* | | *Firn older than 2012 melt layer* | |
| --- | --- | --- | --- | --- |
| | Grain size (mm) | Infiltration ice content (%) | Grain size (mm) | Infiltration ice content (%) |
| | (Mean $\pm$ st. dev.) | | (Mean $\pm$ st. dev.) | |
| 1 | 0.486 $\pm$ 0.141 | 5.94 | 0.697 $\pm$ 0.142 | 29.86 |
| 2 | 0.498 $\pm$ 0.145 | 6.18 | 0.640 $\pm$ 0.144 | 20.41 |
| 3 | 0.484 $\pm$ 0.143 | 5.01 | 0.688 $\pm$ 0.152 | 25.27 |
| 4 | 0.449 $\pm$ 0.111 | 2.19 | 0.651 $\pm$ 0.142 | 21.66 |
| 5 | 0.492 $\pm$ 0.115 | 5.72 | 0.661 $\pm$ 0.146 | 21.74 |
| 6 | 0.426 $\pm$ 0.110 | 2.26 | 0.598 $\pm$ 0.144 | 13.22 |
| 7 | 0.412 $\pm$ 0.068 | 0.07 | 0.530 $\pm$ 0.094 | 1.90 |
| 8 | 0.434 $\pm$ 0.121 | 2.49 | 0.460 $\pm$ 0.126 | 2.88 |
| 10 | 0.382 $\pm$ 0.082 | 0.19 | 0.502 $\pm$ 0.111 | 3.09 |
| 11 | 0.390 $\pm$ 0.091 | 1.57 | 0.501 $\pm$ 0.152 | 3.66 |
| 12 | 0.404 $\pm$ 0.127 | 2.78 | 0.475 $\pm$ 0.132 | 3.17 |
| 13 | 0.445 $\pm$ 0.123 | 2.52 | 0.543 $\pm$ 0.139 | 3.15 |
| 14 | 0.470 $\pm$ 0.140 | 1.21 | 0.491 $\pm$ 0.123 | 1.30 |
| 16 | 0.415 $\pm$ 0.097 | 0.38 | 0.466 $\pm$ 0.113 | 0.38 |

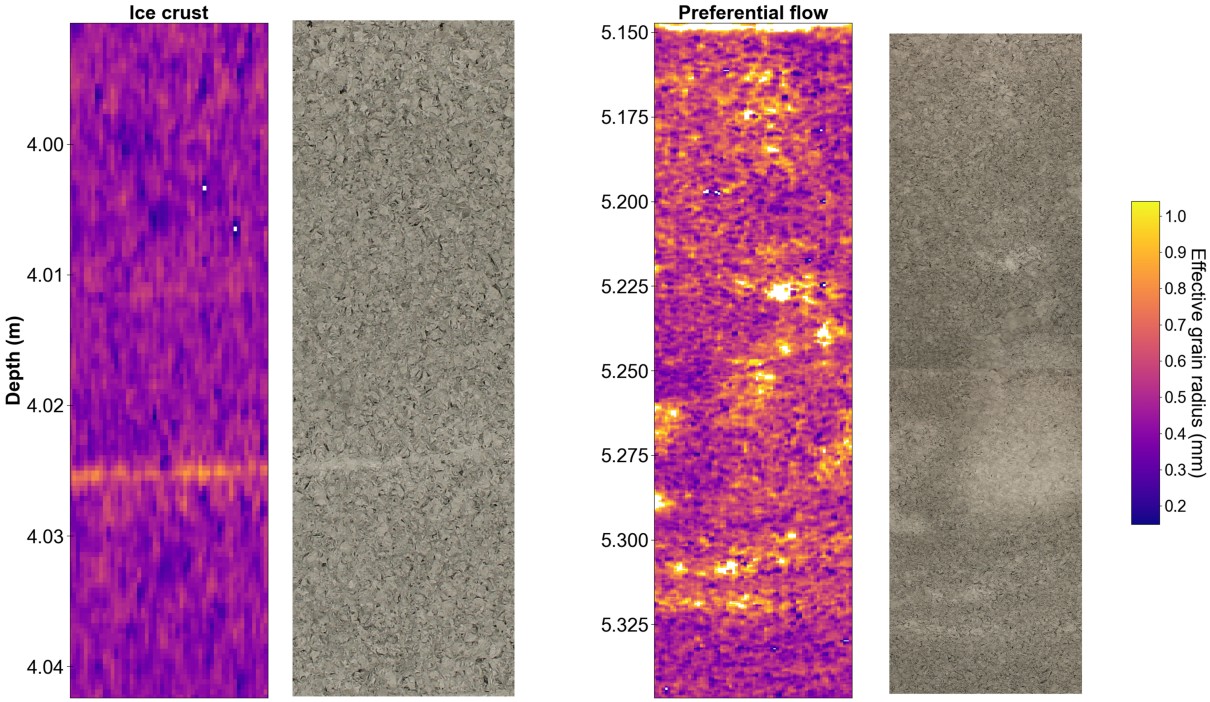

**Figure B1.** Left: Example of a monograin ice crust within Core 10. The ice crust is expressed by larger retrieved grain radii in a narrow band of pixels spanning the core width. A photograph of this feature is shown alongside the hyperspectral map. Right: Example of a preferential flow path found in Core 10. Wet grain growth during preferential flow causes the flow path to be easily detected during grain size retrievals. The two photographs corresponding to this map are stacked and shown on the right. Note the difference in depth scales.

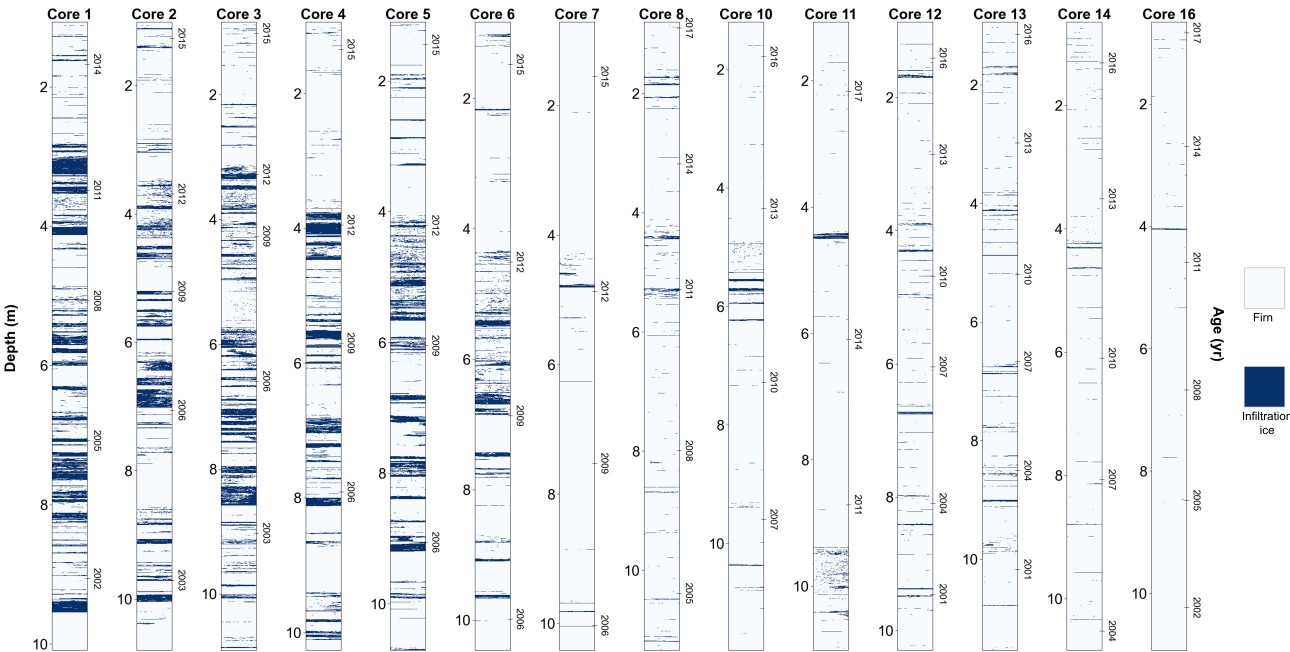

**Figure B2.** Identified infiltration ice within the firn cores from the binarized NIR-HSI grain size maps.

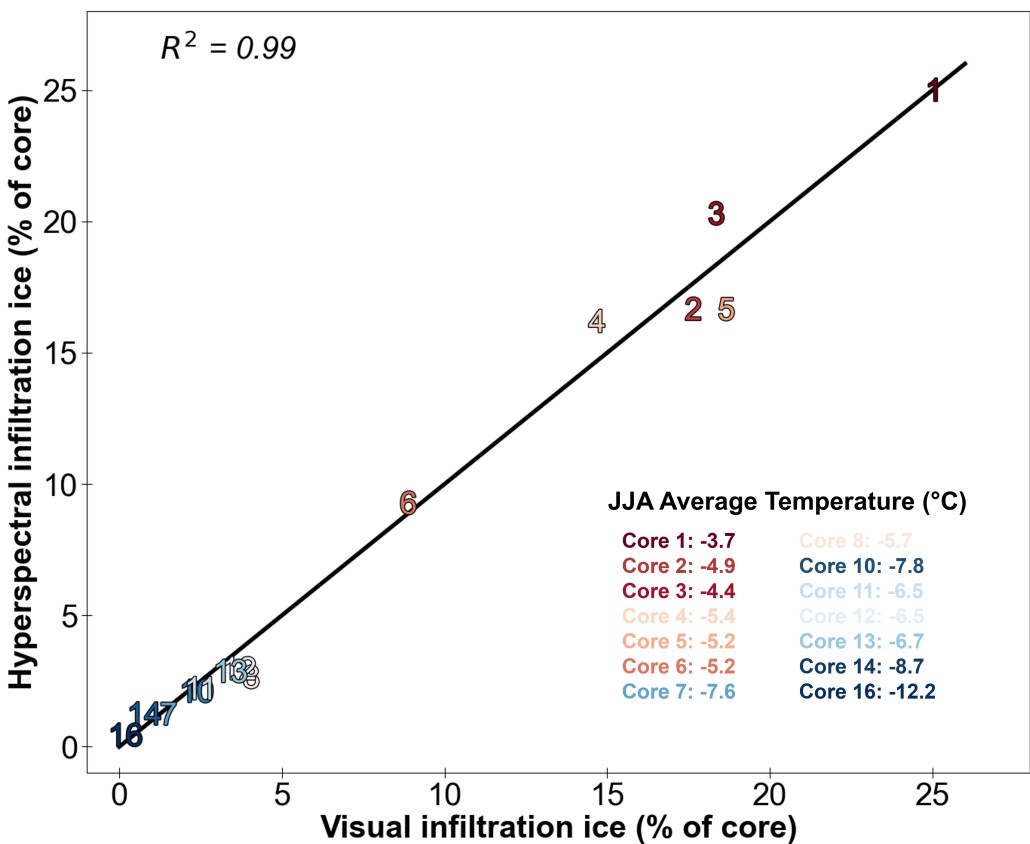

**Figure B3.** Infiltration ice identified in the NIR-HSI scans compared to visually-identified ifiltration ice content in each core. The one-to-one line is shown in black. The core numbers are shaded by summer (JJA) average temperatures estimated by MODIS Land Surface Temperatures.

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
