# Peer review of "A cold laboratory hyperspectral imaging system to map grain size and ice layer distributions in firn cores"

_EGUsphere, 2023_

## Referee Comment (RC1)

**Referee comment to McDowell et al. "A cold laboratory hyperspectral imaging system to map grain size and ice layer distributions in firn cores".**

**General comments**

I enjoyed reading this manuscript by McDowell et al. as it is well written and valuable for the cryo, and especially firn, community. The authors describe a new imaging system based on hyperspectral imaging, which advances the possibilities to map stratigraphy and grain size in firn quickly. The manuscript is well-structured and clearly describes the technique, its results, and its possible limitations. The authors also tackle the tricky question of "what is grain size" and present a well-thought-out approach. I only have a few general thoughts, which are more questions than comments.

I am interested in the total duration of a measurement session, and it would be great if the authors could elaborate on this a bit. How long does the entire procedure take, i.e., preparing the device and firn core, conducting the measurements, and processing the files? What´s the longest you used it in the cold? I guess the question is if the scanner could be used in the field running all day long like a visual stratigraphy line scanner. This would be a real advantage in preventing post-depositional effects and the logistical difficulties of transporting firn cores to the cold lab.

You mention the 16 firn cores drilled, but I would like to know if the two broken cores are important for this study. They are not used, and just ignoring them would increase the readability of the plots.

I only raised a few specific comments below. However, I am confident the authors can provide an updated version for those minor revisions, and I would be happy to see the edited manuscript published in the Cryosphere.

**Specific comments**

L. 1: The phrasing of both ice sheets being "covered in a thick layer of firn" sounds a bit off. The firn thickness of roughly 40-120 m in Greenland and Antarctica is not thick in comparison to the ice sheet thickness. The meaning is clear, but it could be described clearer, e.g. "ice sheets contain a porous layer of firn".

L. 7: "required to test/implement into/check", I am not sure if "to inform" is necessary.

L.8: I see the point that grain size measurements can be subjective, but that depends strongly on the method. Microstructural analyses with e.g. fabric analysers or large area scanning macroscopes of thin and thick sections, respectively, can provide good statistics decreasing the rate of subjectivity. To avoid this issue, you could change it to "time-consuming, and can be subjective depending on the method".

L. 22: I would switch the sentence to "interpreting previous atmospheric compositions via ice cores,…".

L. 27: Matter of definition, but in my opinion, firn belongs to the ice sheets and the firn volume is thus "of" or "within ice sheets".

L. 30: A (half) sentence displaying the processes could help to connect the open porosity of firn fact with the changes in climate and the need for a better understanding, which I totally agree with.

L. 32: Including a new (in review) study could be of interest and would be good to include here to show the state of the art regarding optical methods on firn:

Westhoff, J., Freitag, J., Orsi, A., Martinerie, P., Weikusat, I., Dyonisius, M., Faïn, X., Fourteau, K., and Blunier, T.: Combining traditional and novel techniques to increase our understanding of the lock-in depth of atmospheric gases in polar ice cores - results from the EastGRIP region, EGUsphere [preprint], https://doi.org/10.5194/egusphere-2023-1904, 2023.

L. 39: Again, a question of terminology, but I think "Microstructural properties" would be clearer than "grain-scale properties". Grains could be mistaken for dust grains and thus a different scale.

L. 41: "firn layers" could be confusing here, because it refers to the total firn layer but might also refer to individual layers of firn. "Firn column" is clearer also used in the cited study by Gregory et al. (2014).

L. 68: You include microstructure mapping here, which also works on firn thick sections and is comparably fast and has a very high optical resolution: Kipfstuhl, S., Faria, S. H., Azuma, N., Freitag, J., Hamann, I., Kaufmann, P., Miller, H., Weiler, K., and Wilhelms, F. (2009), Evidence of dynamic recrystallization in polar firn, *J. Geophys. Res.*, 114, B05204, doi:10.1029/2008JB005583.

Recent grain size measurements from ice thin sections via fabric analyser use pixels instead of radii/diameter and are thus able to reproduce a fairly accurate grain area, see e.g. Stoll, N., Eichler, J., Hörhold, M., Erhardt, T., Jensen, C., and Weikusat, I.: Microstructure, micro-inclusions, and mineralogy along the EGRIP ice core – Part 1: Localisation of inclusions and deformation patterns, The Cryosphere, 15, 5717–5737, https://doi.org/10.5194/tc-15-5717-2021, 2021.

Nevertheless, both methods are limited to discrete samples and do not have the advantages of continuous measurements.

L. 72: Baunach et al. 2001 study laboratory grown snow kinetics and measure grain size along the way showing the subjective assessments of six experts. I am not sure if this is a good example to conclude that the described methods above can be subjective. The study is more than two decades old and thus not state-of-the-art any more (as you show with the other cited studies). I am happy to be convinced that this study, and the conclusion you draw, are still as relevant as 2001; some rephrasing might help here. Without a doubt there are enough reasons to develop new methods to measure firn grain size fast and accurately.

L. 74: The topic of the perfect grain size method/tool/parameter has been discussed for decades and there is still no obvious solution due to the 3D shape of grains and the spatial limitation of firn and ice cores. Averaging a large number of grains is thus necessary to obtain "good statistics".

L. 80: You start with "Ice" and then switch to "snow grains". Similar switching occurs in the sentences below. To avoid confusion, it would be good to stick with the same nomenclature of snow/ice grain/particle.

L.. 89: maps of grain size…

L. 90: "in the field" not needed

L. 95-106: This reads more like a summary than the objective paragraph. To be more precise clearly state the objectives of your study here so the reader knows what to expect. It is a good paragraph, just at the wrong location.

L. 109. Maybe directly mention the number of firn cores here. In addition, it would be convenient to state the drilling method (hand-drill, hans-tausen, etc?) without having to read the cited publications.

L. 120: I suspect via commercial companies dealing with frozen goods?

L. 121: …chemical analysis of x using…?

L. 150: Just out of curiosity, what is the maximum time between measurements you left the device in the cold without using it? Could it be insulated to avoid removing after measurements?

L. 179: Nolin Dozier technique (Nolin and Dozier, 2000)

L. 186: Did you play around with the impurity concentration? 0 ppb is very unlikely for natural settings especially in Greenland.

L. 238: Should it be "deep (>10 m)"?

L- 240: number of ice layers

Fig. 3: Do I understand it correctly that you measure effective grain size and then translate those values with a model to radii? So, it is not a direct measurement as one would assume from the figure? If the effective radii are shown please make that clear on the axis label or the caption.

The legend with infiltration ice is slightly confusing, if it refers to the 2012 melt layer, why not mention it here and give it the same black and white line as in the plot.

Fig. 4b: For a more precise comparison, it could be helpful to add the exact mean values for each core next to the dotted line.

L. 284: depth bands could be confusing; I would exchange it to depth regimes.

L. 290: State the three cores here.

L. 312: To demonstrate this point, it would be helpful to see a high-resolution photograph of the characterized infiltration ice. Having a "real" image next to the depth regimes shown in A1 would be great. However, the samples might have been used for other purposes by now so this might not be an option anymore.

Fig. 5 Having a similar plot concept as in Fig. 3 might be more accurate to display the infiltration ice. Now the impression could occur that the percentage/x-axis is the spatial area of infiltrated ice as is the case for visually inspected ice.

L. 319: Is it possible (and maybe even planned) to test the device in the field? Packing, storing and shipping especially of snow and firn is always risky in regards of microstructure so of course it would be great to get these data in the field. The set-up seems to be portable enough to fit into a few Zarges boxes.

Figure 6.a Just to be sure since it is not described, did you mirror the data after measuring the curved site? Some features look mirrored and could explain the visible difference between the left and right side of the core. If it is just a question of lighting, there are probably ways to fix it – how about attaching a strong light source directly to the images as is done for the visual stratigraphy line scanner from e.g. Schäfter+Kirchhoff used by AWI (https://www.sukhamburg.com/products/linescancamera/scannersystems/microstructuremapping/ilcs.html)?

L. 336: Latex format error /sim.

L. 351: It is very reassuring that the focus does not play a major role for grain size analysis. That will make the deployment much easier and less experienced people can easily take over during a measurement campaign. Great that you checked this in advance.

Fig 7b. I see the logic in the arrangement of a, b, and c, but having wider histogram plots in b) would increase the visibility of the two regimes.

L. 402: Since they are labelled Core 1-16, I would write "Core 16".

Figure 8 caption: b) instead of c); 2012 melt layer (pink); standard deviation of/in grain sizes

L. 407: Here you only refer to the 14 undamaged firn cores. It might make sense to exclude the two damaged ones completely and thus have easier labels (Core 1-14).

Fig. A4: The legend for the mean annual air temperature seems to be missing.

---

## Author Comment (AC1)

**Author Response to Referee # 1 – Nicolas Stoll**

**General comments**

I enjoyed reading this manuscript by McDowell et al. as it is well written and valuable for the cryo, and especially firn, community. The authors describe a new imaging system based on hyperspectral imaging, which advances the possibilities to map stratigraphy and grain size in firn quickly. The manuscript is well-structured and clearly describes the technique, its results, and its possible limitations. The authors also tackle the tricky question of "what is grain size" and present a well-thought-out approach. I only have a few general thoughts, which are more questions than comments.

We greatly appreciate your careful consideration of the paper, and we are glad that you found the paper well-structured and well-reasoned. We address each of your comments below. Our responses to your comments are in blue, with any changes made to the paper written in *italics*.

I am interested in the total duration of a measurement session, and it would be great if the authors could elaborate on this a bit. How long does the entire procedure take, i.e., preparing the device and firn core, conducting the measurements, and processing the files? What's the longest you used it in the cold? I guess the question is if the scanner could be used in the field running all day long like a visual stratigraphy line scanner. This would be a real advantage in preventing post-depositional effects and the logistical difficulties of transporting firn cores to the cold lab.

Thank you for this suggestion. We have added a paragraph after line 165 in the previous draft discussing this:

*"Overall, imaging a single core in the cold laboratory took between 2–3 hours. Setting the focus of the NIR-HSI took between 20–30 minutes and required repeated scans to test the minor adjustments of the objective lens. Once the imager had been focused, unpacking and repackaging each firn core segment before and after the scan was the rate limiting step (~10 min). The scanning process itself took ~10 seconds, and each core consisted of approximately 10–15 seconds. Processing the images in the Spectranon Software (applying the dark correction, converting raw data to radiance, transforming radiance to reflectance) required < 5 minutes."*

We also added a sentence at the end of line 190 in the previous draft to note how much time the grain size retrievals required.
*"Because of the high image resolution and large number of pixels, the inversion to retrieve grain radii lasted 5–10 minutes for each core."*

We expound on potential difficulties of use in the field in response to specific comments below. While the imager itself is easily transportable, the biggest concern we have is leaving the imager in the cold for extended periods when not in use without some ability to heat it. See our comments on this below.

You mention the 16 firn cores drilled, but I would like to know if the two broken cores are important for this study. They are not used, and just ignoring them would increase the readability of the plots.

Thanks for raising this point; we appreciate any suggestion to make the paper and figures clearer. While we wish we could get grain size data from all 16 cores, the two missing cores here do not affect any results presented in the paper. We appreciate the suggestion to ignore the missing cores; however, we mention them in the paper because data from these cores have been presented in other publications (e.g., Lewis et al., 2019; Meehan et al., 2021). We wanted to remain consistent with the nomenclature developed during the GreenTrACS study and previously published, which is why we have cores 1-8,10-14, and 16.

Lewis, G., Osterberg, E., Hawley, R., Marshall, H. P., Meehan, T., Graeter, K., ... & Ferris, D. (2019). Recent precipitation decrease across the western Greenland ice sheet percolation zone. *The Cryosphere*, *13*(11), 2797-2815.

Meehan, T. G., Marshall, H. P., Bradford, J. H., Hawley, R. L., Overly, T. B., Lewis, G., ... & McCarthy, F. (2021). Reconstruction of historical surface mass balance, 1984–2017 from GreenTrACS multi-offset ground-penetrating radar. *Journal of Glaciology*, *67*(262), 219-228.

I only raised a few specific comments below. However, I am confident the authors can provide an updated version for those minor revisions, and I would be happy to see the edited manuscript published in the Cryosphere.

Thank you for providing these specific comments below, we believe they improved the presentation of the manuscript.

**Specific comments**
L. 1: The phrasing of both ice sheets being "covered in a thick layer of firn" sounds a bit off. The firn thickness of roughly 40-120 m in Greenland and Antarctica is not thick in comparison to the ice sheet thickness. The meaning is clear, but it could be described clearer, e.g. "ice sheets contain a porous layer of firn".

Thanks for the suggestion to make this sentence clearer. We have removed "thick" from the original sentence. It now reads *"...ice sheets are covered in a layer of porous firn"*. We think this sentence highlights that the firn layer is at the surface of ice sheets and contains open pore space.

L. 7: "required to test/implement into/check", I am not sure if "to inform" is necessary.
We have replaced "to inform" with *"to check"*.

L.8: I see the point that grain size measurements can be subjective, but that depends

strongly on the method. Microstructural analyses with e.g. fabric analysers or large area scanning macroscopes of thin and thick sections, respectively, can provide good statistics decreasing the rate of subjectivity. To avoid this issue, you could change it to "Time consuming, and can be subjective depending on the method".

That is a good point - thanks. We have added this phrasing to the sentence.

L. 22: I would switch the sentence to "interpreting previous atmospheric compositions via ice cores,…".

We have reworded this part of the sentence. It now reads *"...interpreting previous atmospheric compositions using ice cores…"*

L. 27: Matter of definition, but in my opinion, firn belongs to the ice sheets and the firn volume is thus "of" or "within ice sheets".

Good point. We changed "on" to *"of"*.

L. 30: A (half) sentence displaying the processes could help to connect the open porosity of firn fact with the changes in climate and the need for a better understanding, which I totally agree with.

Thanks for the suggestion. The end of this paragraph now reads:

*"The interconnected interstitial spaces between firn grains, i.e., open porosity, allows for gas, vapor, and liquid movement within the column; however, the total open porosity of the firn column is dependent on local climate conditions (e.g., Gregory et al., 2014) and can be progressively reduced by filling with meltwater (e.g., Harper et al., 2012). Therefore, an understanding of firn structure and properties, and their spatiotemporal evolution, is critical to determine how ice sheets respond to changes in climate."*

L. 32: Including a new (in review) study could be of interest and would be good to include here to show the state of the art regarding optical methods on firn:

> Westhoff, J., Freitag, J., Orsi, A., MarAnerie, P., Weikusat, I., Dyonisius, M., Faïn, X., Fourteau, K., and Blunier, T.: Combining traditional and novel techniques to increase our understanding of the lock-in depth of atmospheric gases in polar ice cores - results from the EastGRIP region, EGUsphere [preprint], https://doi.org/10.5194/egusphere-2023-1904, 2023.

Thank you for pointing out this very interesting study. We have included it as a reference here on line 34.

L. 39: Again, a question of terminology, but I think "Microstructural properties" would be clearer than "grain-scale properties". Grains could be mistaken for dust grains and thus a different scale.

Good point. We have changed "Grain-scale properties" to *"Microstructural properties"*.

L. 41: "firn layers" could be confusing here, because it refers to the total firn layer but might also refer to individual layers of firn. "Firn column" is clearer also used in the cited study by Gregory et al. (2014).

*We understand that this could be confusing to readers. The two sentences here now read:*
*"The relationship between gas diffusivity and firn permeability differs depending on firn grain size (Adolph and Albert, 2014) and pore close-off in finer-grained firn layers is reached at shallower depths in the firn column than it is for coarser-grained layers regardless of the density of the layers at depth (Gregory et al., 2014). These grain size effects must be accounted for when determining ice age – gas age differences in ice core records."*

L. 68: You include microstructure mapping here, which also works on firn thick sections and is comparably fast and has a very high optical resolution:

> Kipfstuhl, S., Faria, S. H., Azuma, N., Freitag, J., Hamann, I., Kaufmann, P., Miller, H., Weiler, K., and Wilhelms, F. (2009), Evidence of dynamic recrystallization in polar firn, J. Geophys. Res., 114, B05204, doi:10.1029/2008JB005583.

Recent grain size measurements from ice thin sections via fabric analyser use pixels instead of radii/diameter and are thus able to reproduce a fairly accurate grain area, see e.g.

> Stoll, N., Eichler, J., Hörhold, M., Erhardt, T., Jensen, C., and Weikusat, I.: Microstructure, microinclusions, and mineralogy along the EGRIP ice core – Part 1: Localisation of inclusions and deformation patterns, The Cryosphere, 15, 5717–5737, https://doi.org/10.5194/tc-15-5717-2021, 2021.

Nevertheless, both methods are limited to discrete samples and do not have the advantages of continuous measurements.

*Thanks for these suggested references. This sentence and the following sentence now read:*
*"Firn grain size datasets include "traditional" measurements produced by measuring the largest extent of grains using either a crystal card (e.g., Harper et al., 2003, thin sections (e.g., Gow, 1969; Alley et al., 1982), or digital photographs (e.g., McDowell et al., 2023); outlining grain boundaries in scanning electron microscope (SEM) scans (e.g., Spaulding et al., 2010); extracting grain or crystal boundaries by tracing thermal grooves from optical microscope images of sublimed microtomed thin/thick sections (e.g., Kipfstuhl et al., 2009; Stoll et al., 2021); or calculating the specific surface area in microcomputer tomography (microCT) measurements (e.g., Freitag et al., 2004; Linow et al., 2012). While these methods are time-consuming and tedious, they include additional downsides: measuring traditional grain extents visually can be subjective (e.g., Baunach et al., 2001; Leppänen et al., 2015), while sample preparation for microCT, SEM, and optical microscope analyses is destructive to existing cores and their small size limits their representativeness."*

L. 72: Baunach et al. 2001 study laboratory grown snow kinetics and measure grain size along the way showing the subjective assessments of six experts. I am not sure if this is a

good example to conclude that the described methods above can be subjective. The study is more than two decades old and thus not state-of-the-art any more (as you show with the other cited studies). I am happy to be convinced that this study, and the conclusion you draw, are still as relevant as 2001; some rephrasing might help here. Without a doubt there are enough reasons to develop new methods to measure firn grain size fast and accurately.

While we agree that there have been many advances in grain size observational techniques since 2001, we believe that including this phrase about subjectivity is important. Many grain size measurements from the field are still made by visually estimating the extent of grains. Additionally, because many grain growth parameterizations in snow/firn models were validated using traditional grain size estimates, we think it is important to note their subjectivity here. We have also included the reference to Leppänen et al. (2015), as they show that subjectivity is introduced to grain size estimates from digital photographs of snow grains. See the revised sentence above.

L. 74: The topic of the perfect grain size method/tool/parameter has been discussed for decades and there is still no obvious solution due to the 3D shape of grains and the spatial limitation of firn and ice cores. Averaging a large number of grains is thus necessary to obtain "good statistics".

We agree with this point. This section is not meant to disparage other techniques, but rather to let readers know a gap that the hyperspectral images fill; that is, to provide continuous grain size profiles non-destructively. We have slightly revised the text here to make this point clearer. It now reads:

*"Grain size estimates produced using these techniques are averaged over the sample depth to obtain characteristic statistics and therefore do not produce continuous grain size profiles. Additionally, augmenting these records with ice layer stratigraphy requires visually inspecting firn cores or snowpit walls. These disadvantages motivate the development of a method that can quickly, continuously, and systematically map firn grain size and ice layer stratigraphy."*

L. 80: You start with "Ice" and then switch to "snow grains". Similar switching occurs in the sentences below. To avoid confusion, it would be good to stick with the same nomenclature of snow/ice grain/particle.

We have changed the wording in this paragraph to consistently use "snow grains" wherever we previously had written "ice particles", "ice grains", etc. We do keep "ice" in some places, since it is the absorptive property of ice itself that these techniques are leveraging.

L.. 89: maps of grain size…
Changed. Thanks for catching the typo.

L. 90: "in the field" not needed
Deleted.

L. 95-106: This reads more like a summary than the objective paragraph. To be more precise clearly state the objectives of your study here so the reader knows what to expect. It is a good paragraph, just at the wrong location.

Thanks for raising this point. We agree with you that this paragraph could be more focused to guide readers as to the objectives of this study. The new paragraph now reads:

*"This study was motivated by the need for high-resolution datasets of firn grain size and ice layer stratigraphy for a variety of firn research applications. We aimed to test the performance of a NIR-HSI system in retrieving accurate and continuous grain size profiles and ice layer distributions from 14 firn cores in a cold laboratory. To evaluate the efficacy of the NIR-HSI grain size retrievals, we (1) tested the sensitivity of retrieved effective grain sizes to the orientation of firn cores and the objective lens focus of the NIR-HSI; (2) compared the effective grain size retrievals with "traditional" grain size measurements colocated in 7 cores; amd (3) correlated visual ice layer distributions with ice layer stratigraphy generated by the NIR-HSI. We demonstrate that scanning firn cores with a NIR-HSI is a robust technique for developing detailed grain size and ice layer profiles, and demonstrate an application of the high-resolution dataset to quantify structural changes to the firn column following the extreme 2012 summer melt event."*

L. 109. Maybe directly mention the number of firn cores here. In addition, it would be convenient to state the drilling method (hand-drill, hans-tausen, etc?) without having to read the cited publications.

We have specified that we scanned 14 firn cores in this sentence. Additionally at the end of the paragraph we have added:

*"All cores were collected using a hand auger with a sidewinder attachment and reached depths between 20 - 30 m."*

L. 120: I suspect via commercial companies dealing with frozen goods?

Yes, you are correct. The cores were transported from the field to a -20C freezer in Kangerlussuaq via twin otter aircraft. They were then flown to Schenectady, NY via a New York Air National Guard LC130 aircraft where the cores were loaded into a commercial freezer truck and delivered to Dartmouth College.

We have added *"via a commercial freezer truck"* to this sentence.

L. 121: …chemical analysis of x using…?

We have revised this sentence to now say:

*"...sampled for chemical analysis of water isotopes and major ion concentrations using a continuous melting system…"*

L. 150: Just out of curiosity, what is the maximum time between measurements you let the device in the cold without using it? Could it be insulated to avoid removing after measurements? The maximum time that we left the imager in the cold room was about 2-3 hours (the longest length of time of a measurement session). The manufacturer guidelines recommend using the NIR-HSI in a temperature range of 5–40 C. Donahue et al. (2021, 2022) frequently used the imager in a cold room outside of this temperature range, and when the NIR-HSI was returned to the manufacturer for calibration, they noted that the optics were still aligned so the cold did not affect it. Other researchers have flown this sensor on a drone in the winter time which would also be outside of the manufacturers temperature range and it has performed well. However, we do not recommend leaving it in the cold laboratory full time or for long durations without use because insulating it won't be effective unless it is also heated.

L. 179: Nolin Dozier technique (Nolin and Dozier, 2000)
Fixed.

L. 186: Did you play around with the impurity concentration? 0 ppb is very unlikely for natural settings especially in Greenland.
You are correct that 0 ppb is not realistic for natural settings, but this is an acceptable simplification for grain size forward modeling because impurities lower reflectance primarily in the visible wavelengths. This impact does not extend out into the portion of the NIR spectrum used to retrieve grain size (wavelengths spanning the ice absorption feature centered at 1030 nm). A useful visualization of this can be found in Figure 2a,b from Bohn et al., 2021.

> Bohn, N., Painter, T. H., Thompson, D. R., Carmon, N., Susiluoto, J., Turmon, M. J., ... & Guanter, L. (2021). Optimal estimation of snow and ice surface parameters from imaging spectroscopy measurements. *Remote Sensing of Environment*, *264*, 112613. https://doi.org/10.1016/j.rse.2021.112613

We have added this statement of justification to the text:
*"While an impurity concentration of 0 ppb is not realistic for natural settings, it is an acceptable simplification for this forward modeling because light absorbing particles lower reflectance primarily in the visible wavelengths and this impact does not extend into the portion of the NIR spectrum used to retrieve grain size (Bohn et al., 2021)"*

L. 238: Should it be "deep (>10 m)"?
Yes, thanks for catching this. We have made this correction.

L- 240: number of ice layers
Fixed. Thanks for catching the typo.

Fig. 3: Do I understand it correctly that you measure effective grain size and then translate those values with a model to radii? So, it is not a direct measurement as one would assume from the figure? If the effective radii are shown please make that clear on the axis label or the caption.The legend with infiltration ice is slightly confusing, if it refers to the 2012 melt layer, why not mention it here and give it the same black and white line as in the plot. Apologies for the confusion. This point was also raised by the second reviewer. We have edited this figure to make the legend clearer. We also use *"Effective grain radius"* in our figure labels throughout the paper.

[Figure]

*Figure 5. Firn core stratigraphy maps. (a) Inset map from Figure 1 with firn core locations labeled in black, impermeable ice slab extents in red (MacFerrin et al., 2019), and the 2012 melt layer detections in blue (Culberg et al., 2021). Operation IceBridge (OIB) flightlines analyzed by Culberg et al. (2021) are displayed as thin blue lines. (b) Firn core stratigraphy shaded by effective grain radius ($r_e$). Pixels with an effective grain radius >1.04 mm are classified as infiltration ice and masked. The black and white dashed extent indicators denote firn deposited between January 2011 and January 2013, which should have been affected by the extreme melt event of 2012, that are shown in Figure 8.*

Fig. 4b: For a more precise comparison, it could be helpful to add the exact mean values for each core next to the dotted line.
We have revised this figure in response to a suggestion from Reviewer 2 to examine differences between grains in wetted firn sections and in dry firn sections. We have included the mean

values for the full core grain size comparisons, the wetted firn grain size comparisons, and the dry firn grain size comparisons in each subplot.

[Figure]

***Figure 6.*** *Effective vs. traditional grain sizes. (a) Grain size profiles from digital grain diameter measurements from McDowell et al. (2023) (grey) and from the NIR-HSI (black). Regions of refrozen firn indicating previous wetting from McDowell et al. (2023) are shown in light purple. (b) Ratios of effective grain sizes to traditional grain sizes. Dry firn grain ratios are shown in orange, while firn grains from previously-wetted regions are in purple. The dashed orange and purple lines represent the mean ratios of dry firn grains and wet firn grains respectively. The mean ratio from the full core is shown as a black line. Each ratio for the individual cores are written and colored corresponding to their classification. The light gray shading denotes the range of Effective/Traditional ratios expected given a hypothetical firn grain geometry of a truncated octahedra.*

L. 284: depth bands could be confusing; I would exchange it to depth regimes.
To try to avoid confusion here, we have changed the wording from "depth bands" to *"core segments"*.

L. 290: State the three cores here.

We have replaced "3 of the cores" with *"Cores 3 - 5".*

L. 312: To demonstrate this point, it would be helpful to see a high-resolution photograph of the characterized infiltration ice. Having a "real" image next to the depth regimes shown in A1 would be great. However, the samples might have been used for other purposes by now so this might not be an option anymore.
Thank you very much for this helpful suggestion. We have added photographs of the two regimes next to the hyperspectral maps.

[Figure]

*Figure A2. Left: Example of a monograin ice crust within Core 10. The ice crust is expressed by larger retrieved grain radii in a narrow band of pixels spanning the core width. A photograph of this feature is shown alongside the hyperspectral map. Right: Example of a preferential flow path found in Core 10. Wet grain growth during preferential flow causes the flow path to be easily detected during grain size retrievals. The two photographs corresponding to this map are stacked and shown on the right.Note the difference in depth scales.*

Fig. 5 Having a similar plot concept as in Fig. 3 might be more accurate to display the infiltration ice. Now the impression could occur that the percentage/x-axis is the spatial area of infiltrated ice as is the case for visually inspected ice.
We appreciate your suggestion to improve the clarity of the figure. We have attempted multiple ways to show this comparison. However, we believe that the way it was presented here is the easiest way to visualize the comparison. You do raise an important point about possible

confusion. The x-axis now reads *"% of Pixels Classified as Infiltration Ice in Hyperspectral Images"* and we have re-written the figure caption to read:
*"Comparison of infiltration ice mapped by the NIR-HSI and identified by visual inspection. Profiles from the NIR-HSI are represented as the line-by-line percentage of pixels classified as infiltration ice (black line corresponding to the x-axis). The horizontal and vertical extent of ice layers identified on a light table is shown in blue."*

L. 319: Is it possible (and maybe even planned) to test the device in the field? Packing, storing and shipping especially of snow and firn is always risky in regards of microstructure so of course it would be great to get these data in the field. The set-up seems to be portable enough to fit into a few Zarges boxes.
We don't know of any concrete plans to take this system into glaciological field settings, but you are correct that it would be best to collect these data before any potential microstructural changes occur during transport and storage. The size of the imager certainly makes it easily transportable, and there has been previous work that mounts the imager on drones in cold field settings. The main limitations for a field setup would be creating a dark space with a nadir broadband illumination source (which could be difficult during polar summer in remote field settings) and a mechanism for keeping the imager warm when not in use while it is in the field for many days.
We included  a short discussion of potential uses in the field in our Conclusions section:
*"Given the availability and transportability of NIR-HSIs, these instruments may provide opportunities to map firn microstructure in field settings, which would prevent post-depositional effects on firn structure during transport and storage. However, difficulties in using the NIR-HSI in remote field settings may arise from creating proper illumination conditions and keeping the imager warm when not in use to prevent the optics from becoming misaligned in conditions outside of the recommended range of operating temperatures. However, because firn grain size is important for many ice sheet research applications, we encourage the use of these systems in the lab or possibly in the field to constrain firn grain size across a wide variety of ice sheet settings."*

Figure 6.a Just to be sure since it is not described, did you mirror the data after measuring the curved site? Some features look mirrored and could explain the visible difference between the left and right side of the core. If it is just a question of lighting, there are probably ways to fix it – how about attaching a strong light source directly to the images as is done for the visual stratigraphy line scanner from e.g. Schäter+Kirchhoff used by AWI (https://www.sukhamburg.com/products/linescancamera/scannersystems/microstructurem apping/ilcs.html)?
We have checked this figure and the data have not been mirrored. The differences solely come from centimeter-scale differences in infiltration ice. We agree that mounting a lighting source directly to the imager would partially-remedy the cross-core gradient in grain sizes.  However, we did not have a lighting system like this available to use for this project. Furthermore, a curved

firn core surface would still present difficulties. By imaging a curved surface, not only does the local illumination angle change, but the viewing angle also changes at that point. This has large implications for the magnitude of measured reflectance due to the anisotropic forward scattering of snow/ice grains. If these angles are small, the effects will likely be minimal, but along a curved firn core you approach large grazing angles which would likely produce large differences in reflectance regardless of the light source. We have added to the paragraph originally on lines 334-341:

*"While a nadir light source, e.g., lamps mounted directly around the imager's lens similar to the setup in Donahue et al. (2022), might remedy the cross-core gradient in retrieved grain size, imaging a curved core with the NIR-HSI will still present challenges. The curvature of a firn sample changes the angular distribution of reflected radiation. Along the core edges, the grazing angle of the illumination source becomes large, which will reduce measured reflectance from forward scattering. Therefore, we recommend any detailed analyses of firn structure using the NIR-HSI be conducted on half-round firn cores."*

L. 336: Latex format error /sim.
Fixed. Thanks for catching this Latex error.

L. 351: It is very reassuring that the focus does not play a major role for grain size analysis. That will make the deployment much easier and less experienced people can easily take over during a measurement campaign. Great that you checked this in advance.
Thanks for your positive comment. We agree that it is indeed a nice result that minor changes in focus will not affect results.

Fig 7b. I see the logic in the arrangement of a, b, and c, but having wider histogram plots in b) would increase the visibility of the two regimes.
We agree that the histograms were difficult to see. We have changed these to violin plots and we hope this improves the visibility.

[Figure]

**Figure 8.** *Firn structure within/outside of the 2012 melt layer. (a) Grain size and ice layer maps of firn spanning the 2012 melt layer from 1 September 2012 to 1 January 2011. Pixels with a retrieved effective grain radius >1.04 mm are classified as infiltration ice and masked. The thin bands of smaller effective grain size retrievals are artifacts caused by lighting effects at the ends of firn core segments that have not completely been removed by image cropping. (b) Violin plots showing grain size differences from firn within the melt layer (orange) and outside of the melt layer (purple). Horizontal lines represent the means of the grain size distributions. (c) Bar charts quantifying the amount of infiltration ice found within the 2012 melt layer (orange) and outside of the melt layer (purple). Table A1 contains values of the mean +/- standard deviation grain sizes and total infiltration ice content in firn within and outside of the 2012 melt layer.*

L. 402: Since they are labeled Core 1-16, I would write "Core 16".
Done.

Figure 8 caption: b) instead of c); 2012 melt layer (pink); standard deviation of/in grain sizes
Thanks for catching these typos in the figure caption. These have been fixed.

L. 407: Here you only refer to the 14 undamaged firn cores. It might make sense to exclude the two damaged ones completely and thus have easier labels (Core 1-14).

Thanks for the suggestion to improve clarity. Please see our previous comment on our reasoning behind keeping the original nomenclature from the GreenTrACS study (to remain consistent with other studies presenting data from the same cores).

Fig. A4: The legend for the mean annual air temperature seems to be missing.
Sorry about that. We have added the legend of mean air temperatures. Thanks for catching this! Here is the new figure below:

[Figure]

---

## Author Comment (AC2)

**Author Response to Referee #2 - Anonymous Reviewer**

This manuscript introduces a new near-infrared hyperspectral (NIR HSI) scanning system for fast and continuous laboratory measurements of effective grain radius and infiltration ice content on polar firn cores. The system is based on previous work on seasonal snow and the authors first introduce their modifications to the system for firn core scanning. They then demonstrate that the results are consistent with traditional measurement of grain size and ice layer stratigraphy. They verify that cores should be cut into half rounds before scanning and that the results are largely insensitive to minor changes in objective lens focus. They then discuss the impact of the 2012 extreme melt season on firn structure as manifest in the 14 cores they study.

Overall, this is a well-written paper on an interesting analysis technique that has the potential to greatly improve the availability and resolution of structural data from firn cores, particularly if it can be adapted for use in the field. The paper seems quite technically complete. My only major concerns are with some of the discussion and interpretation of the 2012 melt signatures, but I also have a number of minor comments that I hope can help improve the clarity of the paper.

We would like to thank the reviewer for their thoughtful comments and constructive suggestions that have improved the quality of the paper. We address each of your comments below. Our responses to your comments are in blue, with any changes made to the paper written in *italics*.

**Major Comments:**
[1] How did you choose a threshold grain size of 1mm for classifying infiltration ice? Given you have a visual stratigraphy record for seven of the cores, I would have liked to see a more quantitative optimization of the grain size cutoff used to choose a threshold that maximizes the agreement between the NIR and visual ice layer stratigraphy.
We would like to thank the reviewer for this helpful suggestion. Initially, we chose a threshold radius of 1 mm somewhat subjectively, but we tuned the threshold until there was visually good agreement between visual and hyperspectral ice layer distributions, which we confirmed by showing in Figure A4 that this threshold radius reproduces ~99% of the visual ice layer stratigraphy. However, we agree that this methodology could be more robust, and we have added a subsection in the Methods section describing how we choose an optimal grain radius threshold for masking infiltration ice features, and we now use a threshold grain radius of 1.04 mm to mask infiltration ice in the paper.
*"Pixels containing infiltration ice from refrozen meltwater are immediately apparent in the raw grain size retrievals as anomalously large radii compared to the surrounding firn grains. We developed an effective grain size threshold to classify infiltration ice features in the cores in order to both remove them from the maps to prevent them from biasing average grain sizes, and to create explicit maps of ice layer stratigraphy.*

*We first calculated the total percentage of infiltration identified visually in each core by summing the fractional extent of each ice layer in the core (expressed as a total height of ice) and dividing by the length of the full core (Eq. 2):*

$$VI = 100 \times \frac{\sum (H_n \times W_n)}{L} \qquad (2)$$

*where VI is the percentage of infiltration ice identified through visual inspection, $H_n$ is the thickness of ice layer n, $W_n$ is the width of ice layer n expressed as a fraction of the core width, and L is the total length of the core.*

*We compared VI to the percentage of infiltration ice in the hyperspectral images. To quantify the hyperspectral infiltration ice, HI, we classified pixels as either infiltration ice or firn based on a threshold grain radius (Eq 3):*

$$HI = 100 \times \frac{P_i}{P_t} \qquad (3)$$

*where HI is the percentage of infiltration ice in the hyperspectral images, $P_i$ is the number of pixels classified as infiltration ice, and $P_t$ is the total number of pixels in the full core image.*

*We leveraged our dataset of visual ice layer stratigraphy to determine the threshold grain radius characterizing infiltration ice. We iterated over a range of possible threshold grain radii ranging from 0.7 – 1.3 mm with a step size of 0.1 mm, and categorized any pixel with a retrieved radius greater than the threshold as infiltration ice and calculated HI for each core. We regressed HI versus VI for all cores and calculated the root mean square error (RMSE) during each iteration. Upon finding the threshold radius that minimized the RMSE of the regression (1 mm), we cycled over threshold radii between 0.95 mm and 1.05 mm, increasing the threshold by 0.01 mm for each iteration. Through this process, we minimized the RMSE between the hyperspectral infiltration ice content and the visual ice content and determined a threshold radius with a precision of 0.01 mm. An infiltration ice threshold radius of 1.04 mm minimized the RMSE (1.03%) (Figure 3).*

[Figure]

*Figure 3. The iteration process to determine the threshold grain radius used to classify infiltration ice. (a) A threshold radius of 1.04 mm minimized the RMSE of the regressed hyperspectral infiltration ice versus the visually-detected infiltration ice. (b) Regressions of hyperspectral infiltration ice versus visual infiltration ice for each threshold radius in the iteration are shaded based on the radius used to threshold out infiltration ice. Black line denotes the 1:1 line, and the regression using the 1.04 mm threshold is shown with the red dotted line. Each point represents the infiltration ice content for different cores.*

*We also set a lower-bound radius threshold of 0.15 mm to remove anomalously small grain sizes retrieved along breaks in core segments by by reconciling grain radii maps from the NIR-HSI and visual stratigraphy and identifying regions where grain sizes either sharply decreased at breaks in the core segments."*

[2] GreenTRACS cores 10, 11, 12, 14, and 16 were collected at locations that did not have coincident OIB radar collection in Spring 2017. As a result, the lack of 2012 melt layer detections at these sites in the Culberg et al. (2021) dataset is simply because there was no data in those regions to analyze. This means that most of the discussion from lines 374-382 needs to be removed or rewritten since the absence of the ice layer in those regions does not actually say anything about the detection limits of the radar. On the other hand, Core 1 did in fact have 2017 radar coverage, but the 2012 melt layer was not detected. Culberg et al. (2021) speculated that this might be because there was so much infiltration ice throughout the entire firn column that the 2012 melt layer did not form a unique or distinct ice package with strong density contrasts relative to the surrounding firn. The data presented in this paper seems to me to confirm that speculation quite nicely, and that would be an appropriate comparison and discussion that could be included in the paper.

Thank you very much for this insightful comment. We believe that providing this more nuanced discussion of the relationship to the 2021 paper by Culberg et al. has significantly improved this section of the paper. We have revised this section to now read:

*"Structural changes to the firn column from surface melting in the summer of 2012 were detected in radar soundings from ~9000 km of flight survey lines across all sectors of the Greenland Ice Sheet (Culberg et al., 2021). The lack of flight coverage or radiometric issues with the radar data prevented a melt layer detection at 5 core sites (Cores 10, 11, 12, 14, and 16) (Figure 5a). Given that surface melting occurred across 97% of the Greenland Ice Sheet during this extreme summer (Nghiem et al., 2012), evidence of meltwater percolation should be apparent within the shallow firn column across most of the ice sheet. Our stratigraphic maps provide evidence of a 2012 melt layer that is structurally-different from the surrounding firn column in Cores 10, 14, and 16. Each of these cores have elevated levels of infiltration ice within the 2012 melt layer compared to the surrounding sections of firn, and grain sizes in Core 10's melt layer are much larger than in the surrounding firn (Figure 8, Table A1). While we show evidence of a 2012 melt layer in regions that were not analyzed by Culberg et al. (2021), we also find a thick ice layer from the 2012 melt event in Core 1 that was not detected in the radar data. The detection algorithm of Culberg et al. (2021) was based on firn density contrasts that generate powerful radar reflectors. Culberg et al. (2021) speculated that the absence of the 2012 melt layer at lower elevations in the percolation zone was likely a result of increased background levels of infiltration ice in the firn column that prevented ice layers formed in 2012 from generating a sufficient density contrast. Evidence of the 2012 melt layer in these images while being undetected in the radar data appears to confirm this supposition."*

[3] At lines 394-396, you note that Core 11 shows infiltration ice features outside the temporal range expected for melt effects from 2012 and suggest that this indicates deep preferential infiltration and refreezing. However, it seems to me that these features in Core 11 are more likely to have been formed during the relatively high preceding melt years in 2010 or 2011. There is often significant regional variation in exactly which year in the early to mid-2000s had the highest melt volume for any given site, so it's possible that 2010/2011 is a stronger signal at this location. Core 11 is clearly in a unique region with much higher local accumulation rates than its neighbors, so I would not be surprised if local melt also followed a different pattern. To me, that makes more sense than suggesting that the 2012 melt infiltrated more than 2 m without impacting the firn structure before suddenly leading to large/rapid grain growth at depth (e.g., suggesting somehow no wetting front ever formed, despite high melt volumes).

Again, we thank the reviewer for the insightful comment. We agree that differential melt patterns is a more likely cause for the lack of a 2012 melt signal and the evidence of firn wetting in deeper/older firn. We have added a short discussion on this point in a separate paragraph:

*"Core 11 is the only core in which we find no evidence of previous firn wetting during 2012 (Figures 5b, 8); there are no statistical differences between grain sizes and the amount of infiltration ice in firn possibly affected by 2012 melt. Instead, there is a ~1 m thick section of enlarged firn grains and infiltration ice consistent with previously-wetted firn ~0.5 m below the analyzed region of interest that corresponds to firn deposited in 2009--2010. We suspect that the extreme melt season in 2010 created this structurally-different firn layer (Tedesco et al., 2011).*

*The complete absence of any melt signal from 2012 in Core 11 is likely driven by its unique climate setting. There is a high accumulation region near Core 11 that nearly doubles the average accumulation rate at this site compared to neighboring cores (Lewis et al., 2019). In 2012, the accumulation rate at Core 11 (0.61 mm. w.e. yr⁻¹) was almost 3x that at Core 12 (0.25 mm. w.e. yr⁻¹) and also exceeded the accumulation rate at its southern neighbor, Core 10 (0.53 mm. w.e. yr⁻¹) (Lewis, 2021). Given that Core 11 was collected in a distinctly different accumulation regime, we suspect that local melting at this location is also different from other cores, which could explain the absence of a 2012 melt layer here. The high accumulation rate in 2012 could have prevented snowpack metamorphism accelerated by positive albedo feedbacks that would have driven surface melt at this location (e.g., Tedesco et al., 2011)."*

**Minor Comments:**
How good is the vertical positioning in the reconstructed core-length images, compared to the spatial resolution in the vertical?
Good question. The vertical resolution of the grain size maps is 0.4 mm, while the vertical positioning of features in the maps is accurate to within 2–3 cm. We have added text discussing the uncertainties in feature depth in the Methods section as well as a short section of text in the discussion section. Please see these textual additions to your comments below.

Maybe consider adding a table with all the imaging settings in one place for the reader who is interested in reproducing your setup.
We appreciate this suggestion; however, we believe that an addition of a table with this information is unnecessary. We note the frame rate and integration time in Section 2.2, and these settings are the only ones that the user needs to specify in the Spectranon software.

Is the "grain radius" shown in all the figures the effective grain size ($r\_e$)? Perhaps add that label explicitly on the colorbars.
Thank you for helping to improve the clarity of the figures. Reviewer 1 also raised this point. We have changed "Grain radius" to *"Effective grain radius"* in each figure in the revised paper.

Line 29-31: these two sentences could use a better transition. That is, show the reader how sentence #1 implies or leads to the statements in sentence #2.
Thanks for this suggestion. Taking into accounts from both you and Reviewer 1, the end of this paragraph now reads:
*"The interconnected interstitial spaces between firn grains, i.e., open porosity, allows for gas, vapor, and liquid movement within the column; however, the total open porosity of the firn column is dependent on local climate conditions (e.g., Gregory et al., 2014) and can be progressively reduced by filling with meltwater (e.g., Harper et al., 2012). Therefore, an understanding of firn structure and properties, and their spatiotemporal evolution, is critical to determine how ice sheets respond to changes in climate."*

Line 32: should be "pore close-off" rather than "pore-close off" as currently written.
Thanks for catching this. We have fixed this typo.

Line 34: I am not sure if listing past accumulation rates as something that can easily be obtained from density profiles is quite accurate, since core dating is also required, which is a much more complicated/laborious process. But based on the references, maybe you meant that density profiles enable accumulation estimates from ice-penetrating radar measurements? In that case, I would suggest being precise about that application in the writing.
Thanks for raising this point and the suggestion to improve clarity. We have revised this portion of the sentence to say that density can be used to *"ascertain past accumulation rates from ice-penetrating radar measurements (e.g., Miege et al., 2013; Hawley et al., 2014; Lewis et al., 2019)"*.

Line 38: perhaps "not that only important metric to characterize firn structure" would be more correct than "not a perfect proxy for firn structure"? Density certainly is part of the structure, not just a proxy, but it does not tell us all the information we need for sure.
This is a fair point. We have revised the text here to read: *"...indicating that density is not the only metric that should be used to characterize firn structure".*

Lines 43-45: grain size is largely irrelevant for radar measurements since the grain scale is orders of magnitude smaller than the wavelength (mm grain sizes vs decimeter to meter wavelengths). It may be relevant for some high-frequency microwave applications (scatterometers, radiometers, SAR, or radar altimeters) operating at center frequencies greater than 10 GHz. I would drop reference to radar and perhaps add a caveat on the frequency range of interest to the statement of on microwave sensitivity.
Thank you for this comment. We have removed reference to radar and now focus only on microwave/optical remote sensing measurements. The two sentences here are now: *"Additionally, analysis of microwave and optical remotely-sensed measurements to determine ice sheet mass balance requires consideration of firn grain size. The firn's grain size controls the penetration depth of high-frequency (>10 GHz) microwaves in firn and governs scattering and emissivity (Rott et al., 1993; Brucker et al., 2010)."*

Line 89: "maps to millimeter – centimeter scale" is a bit ambiguous. Do this mean that pixel sizes in the image are of this order, or that mm-cm scale grains can be resolved?
Thanks for catching the ambiguity here, and apologies for the confusion. In a previous draft of the manuscript we discussed more background about spectral methods to determine snow grain scale. Airborne spectrometers can map effective grain size across landscapes, but their spatial resolution is on the meter scale. For example, a single grain size is estimated for a ~100 $m^2$ pixel based on the "bulk" spectral reflectance. Contact spectrometers are effective at producing localized grain size estimates (e.g., on a snowpit wall), and their resolution is ~ 1-5 cm. The NIR-HSI can further increase the resolution to the mm scale. For clarification the end of this revised paragraph now reads:

*"The millimeter – centimeter pixel resolution of near-infrared hyperspectral imagers (NIR-HSI) produces grain size maps at a high spatial and spectral resolution. Studies utilizing a NIR-HSI have been proven to efficiently and accurately produce high-resolution maps of grain size of laboratory snow samples (Donahue et al., 2021), along the vertical profile of a snowpit wall (Donahue et al., 2022), and at the snow surface when mounted to a drone (Skiles et al., 2023). In addition to providing grain size data, regions of low reflectance in the high-resolution images can allow for ice layers to be easily detected (Donahue et al., 2021)."*

Lines 100-104: these sentences would be more appropriate for the conclusion than the introduction. Here you just want to present a concise outline of the structure of the paper, not get into discussing the results.

We appreciate the suggestion from both reviewers to outline the structure of the paper rather than provide conclusions here. The revised paragraph now reads:

*"This study was motivated by the need for high-resolution datasets of firn grain size and ice layer stratigraphy for a variety of firn research applications. We aimed to test the performance of a NIR-HSI system in retrieving accurate and continuous grain size profiles and ice layer distributions from 14 firn cores in a cold laboratory. To evaluate the efficacy of the NIR-HSI grain size retrievals, we (1) tested the sensitivity of retrieved effective grain sizes to the orientation of firn cores and the objective lens focus of the NIR-HSI; (2) compared the effective grain size retrievals with "traditional" grain size measurements colocated in 7 cores; amd (3) correlated visual ice layer distributions with ice layer stratigraphy generated by the NIR-HSI. We demonstrate that scanning firn cores with a NIR-HSI is a robust technique for developing detailed grain size and ice layer profiles, and demonstrate an application of the high-resolution dataset to quantify structural changes to the firn column following the extreme 2012 summer melt event."*

Figure 2: consider adding a label for the Spectralon panels.

We have now added labels on the photograph pointing to the spectralon panels.

[Figure]

Lines 180-181: define "continuum normalized absorption feature" and "continuum reflectance" for the reader.

We have slightly reworded this section to provide clarity with those less-familiar with the technique. These lines are now written as:

*"The scaled band area, $A_b$, is the area between the measured reflectance, $R_m$, and the continuum reflectance, $R_c$, integrated over the ice absorption feature centered at 1030 nm and scaled by the continuum reflectance:*

$$A_b = \int_{962\,nm}^{1092\,nm} \frac{R_c - R_m}{R_c}$$

*$R_c$ represents the reflectance spectrum in the absence of ice absorption and is defined as the slope between the shoulders of the ice absorption feature."*

Line 186: did you run any sensitivity tests with higher impurity concentrations? 0 ppb seems quite unlikely for Greenland cores, and I assume there might be some chemical information available from the GreenTRACS cores to inform a better average value.

You are correct that 0 ppb is not realistic for natural settings, but this is an acceptable simplification for grain size forward modeling because impurities lower reflectance primarily in the visible wavelengths. This impact does not extend out into the portion of the NIR spectrum

used to retrieve grain size (wavelengths spanning the ice absorption feature centered at 1030 nm). A useful visualization of this can be found in Figure 2a,b from Bohn et al., 2021.

Bohn, N., Painter, T. H., Thompson, D. R., Carmon, N., Susiluoto, J., Turmon, M. J., ... & Guanter, L. (2021). Optimal estimation of snow and ice surface parameters from imaging spectroscopy measurements. *Remote Sensing of Environment*, *264*, 112613. https://doi.org/10.1016/j.rse.2021.112613

We have added this statement of justification to the manuscript text:
*"While an impurity concentration of 0 ppb is not realistic for natural settings, it is an acceptable simplification for this forward modeling because light absorbing particles lower reflectance primarily in the visible wavelengths and this impact does not extend into the portion of the NIR spectrum used to retrieve grain size (Bohn et al., 2021)"*

Lines 197-198: how do you handle core gaps and gaps from image cropping when reconstructing the full-length core images?
Good question. After cropping ~1 cm off the top and bottom of each image, we stacked them to create an image of the full core and then assigned a depth for each row of pixels by creating a depth array that was the same size as the number of rows and started at the top depth of the core and ending at the bottom depth. The effect of this treatment is that the core image was a few cm shorter than the real core but was then slightly stretched when we assigned the depth array. This certainly could introduce ~2-3 cm uncertainty in the depths of features in the hyperspectral images. However, we also note that determining the visual depths/lengths of cores can have similar uncertainties, since it is often difficult to see where ice layers begin/end and the edges of core segments can be jagged or deteriorated and it is hard to accurately assign a precise start/end depth for each core segment on the light table. For this reason, it is often difficult to get features from the cores to exactly align (as you note in Figure 5).

While this treatment of stacking/stretching core images does not impact depth uncertainty at levels greater than from visual inspection, we do believe that you raise an important point that should be mentioned in the text here. We have added the following text to the end of the paragraph:

*"Once stacked the full image of the core was ≤10 cm shorter than the real core, an artifact of the edge cropping of individual images. We assigned a depth array equal to the number of rows of pixels in the image with the start and end depth equal to the visually-identified top and bottom depths of the core. In effect, by equating the depths this slightly stretched the core image, and the depth uncertainty of features in maps created from the images ~2–5 cm. We note that this is approximately the same as the uncertainty in the depths of visually-identified features, since it is often difficult to measure exactly where ice layers begin/end, and the top and bottom of core segments are often jagged or deteriorated, which makes it challenging to accurately set the*

*length of each core segment. These uncertainties can result in slight depth discrepancies between visually-identified and hyperspectrally-retrieved firn core stratigraphy."*

Section 3.1: is there any relationship between ratio of traditional to effective grain size and the degree of past firn wetting? Squinting at Figure 4, it seems like there might be a larger difference in grain size in wetted regions of the cores, but it's hard to tell. This would be interesting to check in case rapid grain growth in saturated firn produces a different dominant grain geometry than the slower growth in dry firn.

Thank you for this nice suggestion. We have added regions of wetted firn identified through visual inspection on the light table (see McDowell et al. (2023) describing this). We then calculate the traditional/effective ratio for each full core, wetted firn sections, and dry firn sections. The ratios of traditional/effective grain size is actually *smaller* than in wetted firn than in dry firn, suggesting that these wetted grains are rounder than dry firn grains. As Figure 4 from Brun (1989) shows, grains quickly lose their dendritic characteristics and become round as the liquid water content increases. Therefore, wet firn metamorphism will transform these grains into a shape that more-closely resembles an effective sphere.

Brun, E. (1989). Investigation on wet-snow metamorphism in respect of liquid-water content. Annals of glaciology, 13, 22-26.

McDowell, I. E., Keegan, K. M., Wever, N., Osterberg, E. C., Hawley, R. L., & Marshall, H. P. (2023). Firn Core Evidence of Two-Way Feedback Mechanisms Between Meltwater Infiltration and Firn Microstructure From the Western Percolation Zone of the Greenland Ice Sheet. *Journal of Geophysical Research: Earth Surface*, *128*(2), e2022JF006752.

[Figure]

***Figure 6.*** *Effective vs. traditional grain sizes. (a) Grain size profiles from digital grain diameter measurements from McDowell et al. (2023) (grey) and from the NIR-HSI (black). Regions of refrozen firn indicating previous wetting from McDowell et al. (2023) are shown in light purple. (b) Ratios of effective grain sizes to traditional grain sizes. Dry firn grain ratios are shown in orange, while firn grains from previously-wetted regions are in purple. The dashed orange and purple lines represent the mean ratios of dry firn grains and wet firn grains respectively. The mean ratio from the full core is shown as a black line. Each ratio for the individual cores are written and colored corresponding to their classification. The light gray shading denotes the range of Effective/Traditional ratios expected given a hypothetical firn grain geometry of a truncated octahedra.*

Figure 3: consider adding the 2017 CReSIS OIB flight lines in light grey as an underlay in panel (a) so that it is clearer whether gaps in the ice slab and 2012 melt layer detections are due to lack of detections or lack of radar data.

(b)The legend is a bit confusing here, since you use a black box outline to show the 2012 melt layer region, but the legend labels a white box with black outline as being infiltration ice. I assume the legend is trying to indicate that white colors in the grain size colormap are infiltration ice? Maybe it would be better to add white the top of the colorbar, label it as grain size > X, and then just note in the caption that regions of the core where the grain size exceed X are interpreted as infiltration ice.

Thanks for these good suggestions. We have added OIB flightlines analyzed by Culberg et al. (2021) in panel (a) and we have revised the legend per your suggestions. The revised figure is below.

[Figure]

*Figure 5. Firn core stratigraphy maps. (a) Inset map from Figure 1 with firn core locations labeled in black, impermeable ice slab extents in red (MacFerrin et al., 2019), and the 2012 melt layer detections in blue (Culberg et al., 2021). Operation IceBridge (OIB) flightlines analyzed by Culberg et al. (2021) are displayed as thin blue lines. (b) Firn core stratigraphy shaded by effective grain radius ($r_e$). Pixels with an effective grain radius >1.04 mm are classified as infiltration ice and masked. The black and white dashed extent indicators denote firn deposited between January 2011 and January 2013, which should have been affected by the extreme melt event of 2012, that are shown in Figure 8.*

Figure 5: there are some interesting spatial offsets here – for example, Core 10 where the NIR stratigraphy seems to be consistently translated downwards by a centimeter or two compared to the visual stratigraphy. What is the uncertainty in vertical positioning for each of the stratigraphic measurement methods and how might that affect your comparison here?

Thanks for raising this point. It is important to mention the slight offsets arising from depth uncertainty of these features in the text here. We have added the following text to this section to address this. It references our discussion of uncertainty that we have added in the Methods section.

*"There are slight vertical offsets between infiltration ice identified in hyperspectral images and on the light table. These slight discrepancies arise from small (cm) uncertainty in the depths of these features introduced during both the cropping of hyperspectral images and logging core stratigraphy on a light table as described in Section 2.4.1."*

Section 3.3: I would consider moving this section earlier in the results. The best organizational flow would be to first present evidence that your methods are robust (sensitivity tests on core curvature and focusing), then quantitatively verify your results (comparison to traditional grains size and visual stratigraphy), and finish with interpreting those results (2012 melt layer stuff).

Thanks for the helpful organizational suggestion. We agree that is a logical flow to the manuscript. The revised order of the sections is:
3.1. Sensitivity tests
3.2. High resolution maps of grain size
3.3. Traditional and effective grain size comparisons
3.4. 2012 melt layer analysis

Line 336: typesetting issue with the ~
Fixed - thanks for catching this Latex error.

Figure 7: what are the distinct lenses/stripes of low grain radius – for example, in Core 11? Are these physical features, or an effect of core breaks and image splicing?

This is an artifact from lighting variations at the end core segments that were not fully removed by image cropping. Cropping the images required balancing some lighting artifacts to not crop a significant portion of the core. We made a note of this in the figure caption (see below). We also now write on line 197 of the previous version of the manuscript:

*"Some bands of anomalously small retrieved grain sizes appear in the maps that have not been removed from cropping, since we attempted to strike a delicate balance between removing lighting artifacts and not cutting a significant portion of the firn core from the images."*

The histograms in panel b are pretty hard to read with this aspect ratio. It is nice to have them aligned with the cores, so I do not have an immediate easy fix, but consider playing around with some different layouts that might allow for some stretching of the x-axis so that the differences between histogram peaks within each plot are more legible.

We agree that the histograms were difficult to see. We have changed these to violin pots and we hope this increases the legibility of the figure.

[Figure]

**Figure 8.** *Firn structure within/outside of the 2012 melt layer. (a) Grain size and ice layer maps of firn spanning the 2012 melt layer from 1 September 2012 to 1 January 2011. Pixels with a retrieved effective grain radius >1.04 mm are classified as infiltration ice and masked. The thin bands of smaller effective grain size retrievals are artifacts caused by lighting effects at the ends of firn core segments that have not completely been removed by image cropping. (b) Violin plots showing grain size differences from firn within the melt layer (orange) and outside of the melt layer (purple). Horizontal lines represent the means of the grain size distributions. (c) Bar charts quantifying the amount of infiltration ice found within the 2012 melt layer (orange) and outside of the melt layer (purple). Table A1 contains values of the mean +/- standard deviation grain sizes and total infiltration ice content in firn within and outside of the 2012 melt layer.*

Line 432: should be "SMS" not "SMK"?
Thanks for catching this! We have fixed this typo.

---

## Editor Decision (ED1)

Dear Authors

Thank you for your revised version, which I have read in great detail. I am very satisfied with your very careful consideration of the reviewers' comments and for your thorough revisions. Your paper is a useful contribution to firn microstructural characterization and I expect I will soon be able to accept it. Before I do this, however, I would like you to consider the following comment and other minor comments.

An important part of your discussions deals with the comparison of traditional and NIR grain size. As detailed in the comments below, I am not fully convinced by your geometric considerations to explain the larger NIR grain size. I would like you to consider that NIR probes only ice-air interfaces, while in firn a large fraction of grain perimeters are ice-ice interfaces, in fact grain boundaries, which NIR does not detect. Therefore, the lower surface to volume ratio yielded by NIR leads to increased apparent grain size. This artefact is much less important in most snow types where grain boundaries are a much smaller fraction of grain perimeters. I therefore think that interpreting NIR reflectance as grain size is not adequate, as this implicitly implies that there are no grain boundaries. I would rather recommend interpreting NIR reflectance in terms of specific surface area (SSA), which quantifies the air-ice interface without any implicit, and arguably erroneous, hypothesis on grains size or shape. I am fully aware that this may not be the current standard in the field, but I think concepts must evolve as investigating techniques improve and produce different information that should not be interpreted in terms of previous concepts. Please take the time to reflect on this, I am open to discussion.

Please also consider the minor comments below. Line numbers are those in the tracked changes version.

Line 13. I suggest adding « effective » to grain size here since this is now what you produce.

Lines 54-55. "Changes in grain size also create differential forward scattering of green light used in laser-altimetry surveys, which can introduce elevation biases by delaying photon returns to the altimeter (Smith et al., 2018)."
This process takes place in the top snow layer, about the top 10 cm, and not in the firn. You are not studying this layer and your results are therefore not relevant to this problem. I recommend deleting this. You can readily simulate the irradiance profile at 500 nm using TARTES https://snow.univ-grenoble-alpes.fr/snowtartes/index to realize that with about 10 ppb of soot, the e-folding depth is around 10-15 cm, depending on the snow properties you choose (density and SSA).

Line 63. "altimetry-based mass balance assessments". Likewise, if you are referring to the determination of the level of the surface by visible radiation, ice layers will only affect this determination if they are near the surface. If you are now referring to radar, please specify to avoid confusion.

Line 137. How about specifying how thick those ice slabs are?

Line 206. "the the"

Line 212. Eq.1. You therefore only use the data between 962 and 1092 nm. Why then scan the whole 900-1700 nm range? By the way you could also fit the whole spectrum to determine the grain size, for example using TARTES mentioned above. You could easily try it. I am pretty sure it would be at least as good as using the Nolin-Dozier method. That method is good if you only have NIR data, I guess if only a Si photodiode is available, which was probably the case for Nolin and Dozier. If you have an InGaAs photodiode that yields SWIR data to 1700 nm, I am really not sure the Nolin-Dozier method is the best. You do not need to address this is your revision. I am just bringing this to our attention for future research.

Line 212. "in the absence of ice absorption". I am not sure this is the best wording. Ice absorbs significantly at all NIR and SWIR wavelengths. You probably mean "if the 1030 nm band were removed"? Please clarify.

Line 215. How are your grain size results dependent on the SNICAR optical parameterizations, i.e., on the g and B values? You may compare the SNICAR values to those in Robledano et al. (2023) https://doi.org/10.1038/s41467-023-39671-3 and test whether the latest research would change your final results.

Line 218 "Snow density negligibly affects snow reflectance". This is only true for a semi-infinite homogeneous snow layer. In fact, it has no effect. Anyway, your sentence is correct in your context, since your core thickness is probably at least 3 times the e-folding depth, but the statement is not generally true as suggested by your writing.

Line 268 "by by"

Figure 5a. The OIB flightlines appear black to me, not blue.

Lines 371-372. Both Lehning 2002 and Vionnet 2012 have produced models for seasonal snow, not for firn. The physical processes determining grain growth are different. References for firn models would be more appropriate.

Figure 6b. I really have trouble telling the purple from the black in the numbers at the top. More contrasted colors may be useful. I am not color-blind, by the way.

Lines 379-383. This is not totally correct. You are mixing up snow and firn metamorphism, where processes are very different. In firn, the large radii of curvature mean that vapor diffusion is not always predominant, and surface diffusion can become important. Please see e.g., Maeno and Ebinuma (1983) doi:10.1021/j100244a023. There are several other references on the topic.
The temperature gradient in firn is too low to produce faceted crystals. The reference to Fierz 2009, which is for seasonal snow, is not adequate. This paragraph and the next one could probably be condensed to retain only the aspect actually relevant to your study. I suggest just focussing on firn data and processes. You may then, and separately, extend this to a snow discussion, but such a snow discussion does not seem useful to me. I however let you decide on this last point, but please remember that, for a scientific paper, the shorter, the better.

Lines 398-421. Honestly, I am not too thrilled by your explanation of the grain size difference, as mentioned above. First of all, is there any actual observational evidence of faceted forms in firn? I have not seen any but am open to evidence. The evidence you propose seems to be just a modeling choice that has no real basis, if I understand correctly. There is of course the obvious fact that traditional grain size uses a section where grains are not always cut in their center, therefore showing a smaller section. NIR reflectance on the other hand, penetrates several grains thick, so that it probes to a depth of several grains. As detailed in the start of my comments, I suggest you consider that NIR probes air-ice interfaces only, and does not see grain-grain boundaries, while traditional grain size does consider these boundaries. Since grain boundaries make up a significant fraction of a grain perimeter in snow, NIR will inevitably produce a negative artifact intrinsic to the method. If you agree to this, you may mention it in your conclusion. And by the way, I then think that translating NIR reflectance in terms of specific surface area (SSA) rather than grain size would be much more physically meaningful. Grain size implicitly implies that the whole perimeter of a grain is an ice-air interface, which is not true. SSA makes no assumption. The principle of the method you use to determine grain size was developed for snow, where most of the perimeters are indeed ice-air interfaces. Now that you are moving to firn, I extremely strongly recommend that you adjust your interpretation to the reality of that new medium.

Line 415. "The the"

Line 437. I am not sure Fierz 2009 applies here.

Lines 438-441. Would not this insertion be better placed together with the previous insertion? Your decision.

Figure A4. Why use the mean annual temperature? Would not the summer temperature be more appropriate? Or even just the July-August mean temperature? Only periods when melting may take place are relevant, it seems.

Line 520. Should you refer the reader to Figure 5 here? Your call.

Lines 541-542. Are units correct here? Or m rather than mm? No dot please, it is a unit.

Lines 534-545. I am not really convinced by your explanation of the absence of a melt layer in core 11. Do you mean that the high accumulation rate would have decreased the temperature gradient, which would have reduced surface grain growth and therefore increased albedo? This dos not seem to be written very clearly. Other factors would then come into play. For example, more light snow of lower thermal conductivity will reduce downward heat loss during the warm spell and lead to greater heating of surface snow. This effect would be opposite to the one you propose. A quantitative assessment including all energy-relevant processes would be required to reach the conclusion you propose, if I understand it correctly.
How about that the different snow structure, likely lower density, would have favored preferential flow rather than matrix flow, so that some areas would be minimally affected by wetting? Core 11 could then have been drilled in a little-affected spot. Just a thought, your decision. Your subsequent paragraph in fact almost leads to this same suggestion.

---

## Author Response (AR2)

Dear Authors,

Thank you for your revised version, which I have read in great detail. I am very satisfied with your very careful consideration of the reviewers' comments and for your thorough revisions. Your paper is a useful contribution to firn microstructural characterization and I expect I will soon be able to accept it. Before I do this, however, I would like you to consider the following comment and other minor comments.

Dear Dr. Domine,

Thank you for carefully reading our paper. We agree with many of your comments and suggestions, and we have edited the manuscript accordingly. We believe that your comments have further improved the quality of the paper. Please see our responses in blue text, with changes that we have made to the text of the paper in *italics*. Additionally, we have uploaded the revised manuscript using the "Track Changes" function so that our edits will be readily apparent to you.

An important part of your discussions deals with the comparison of traditional and NIR grain size. As detailed in the comments below, I am not fully convinced by your geometric considerations to explain the larger NIR grain size. I would like you to consider that NIR probes only ice-air interfaces, while in firn a large fraction of grain perimeters are ice-ice interfaces, in fact grain boundaries, which NIR does not detect. Therefore, the lower surface to volume ratio yielded by NIR leads to increased apparent grain size. This artefact is much less important in most snow types where grain boundaries are a much smaller fraction of grain perimeters. I therefore think that interpreting NIR reflectance as grain size is not adequate, as this implicitly implies that there are no grain boundaries. I would rather recommend interpreting NIR reflectance in terms of specific surface area (SSA), which quantifies the air-ice interface without any implicit, and arguably erroneous, hypothesis on grains size or shape. I am fully aware that this may not be the current standard in the field, but I think concepts must evolve as investigating techniques improve and produce different information that should not be interpreted in terms of previous concepts. Please take the time to reflect on this, I am open to discussion.

We appreciate your perspective and recognize that large fractions of firn grains may have grain boundaries. We believe that due to the retrieval method effective grain size is the preferred and most straightforward way to define the parameter we present based on the retrieval, and has a solid optical basis and a long legacy in the literature. Both SSA and effective grain size are ways to indicate ice absorption/path length through particles, and there is a well-defined relationship between the two metrics - so translating to SSA would be possible (as we note in the Introduction). However, because the modeling is based on a collection of spherical particles that reflect the same as was measured, the effective grain radius is a more accurate descriptor of the retrieval.

This method has been applied to snow, firn, and ice (e.g. Donahue et al., 2023, Cook et al., 2020, Bohn et al., 2021, Skiles et al., 2023, Negi and Kokhanovsky, 2011) with the understanding that effective grain size is not directly related to physical/observable grain size, which we do explicitly recognize in the paper (Line 343: *Effective and traditional grain sizes are not expected to be the same.*) The relationship between the two, as presented in this paper, is still interesting and worthwhile to present because the two measurements are available for the same cores, and could be useful as an empirical conversion.

We have added the following text to the end of Section 2.3:

*"We directly report the $r_e$ values retrieved through this forward modeling approach, but reiterate that this is an optical property; because of the field of view, and mm-to-cm penetration of NIR light, the resultant pixel reflectance can represent light interactions with multiple grains."*

We do agree with your point that differences between effective and traditional grain sizes likely arise because of the differences in measurement techniques. The NIR-HSI is measuring ice absorption, while a "traditional" measurement is made by measuring observable individual grains so it is understandable that these differences could lead to a high bias in effective grain radius. In response to your additional comments below, we have simplified the discussion of the effective/traditional differences in the main text and moved the grain geometric correction to the appendix.

Please also consider the minor comments below. Line numbers are those in the tracked changes version.

Line 13. I suggest adding « effective » to grain size here since this is now what you produce.

We have updated this sentence to now read: *"We leverage the relationship between effective grain size, a measure of absorption by ice grains, and near-infrared reflectance to produce high-resolution (0.4 mm) maps of effective grain size and ice layer stratigraphy."*

Lines 54-55. "Changes in grain size also create differential forward scattering of green light used in laser-altimetry surveys, which can introduce elevation biases by delaying photon returns to the altimeter (Smith et al., 2018)." This process takes place in the top snow layer, about the top 10 cm, and not in the firn. You are not studying this layer and your results are therefore not relevant to this problem. I recommend deleting this. You can readily simulate the irradiance profile at 500 nm using TARTES https://snow.univ-grenoble-alpes.fr/snowtartes/index to realize that with about 10 ppb of soot, the e-folding depth is around 10-15 cm, depending on the snow properties you choose (density and SSA).

Thank you for providing us with the link to this useful tool. Firn can be exposed at the surface of some alpine glaciers and at low elevation regions of Greenland's percolation zone according to

our definition of firn, which is most commonly used by researchers studying ice sheet mass balance (The Firn Symposium Team, 2024). Where firn is exposed, firn properties may affect optical scattering in these locations, which also happens to be places where glacier/ice sheet elevation changes are large. However, we do recognize that these impacts are not widespread across most of the ice sheet. Because this sentence does not change our study's motivation, results, or conclusions, we have removed the sentence to prevent any confusion from readers.

Line 63. "altimetry-based mass balance assessments". Likewise, if you are referring to the determination of the level of the surface by visible radiation, ice layers will only affect this determination if they are near the surface. If you are now referring to radar, please specify to avoid confusion.

We apologize for the confusion; thanks for requesting the clarification. We have changed this phrasing to *"mass balance assessments from microwave radar surveys"*.

Line 137. How about specifying how thick those ice slabs are?

This sentence now reads *"... and zones with thick (≥ 1 m) ice slabs…"* as this is how ice slabs are defined by MacFerrin et al. (2019).

Line 206. "the the"

Thanks for catching this typo. We have removed the duplicated word.

Line 212. Eq.1. You therefore only use the data between 962 and 1092 nm. Why then scan the whole 900-1700 nm range? By the way you could also fit the whole spectrum to determine the grain size, for example using TARTES mentioned above. You could easily try it. I am pretty sure it would be at least as good as using the Nolin-Dozier method. That method is good if you only have NIR data, I guess if only a Si photodiode is available, which was probably the case for Nolin and Dozier. If you have an InGaAs photodiode that yields SWIR data to 1700 nm, I am really not sure the Nolin-Dozier method is the best. You do not need to address this in your revision. I am just bringing this to our attention for future research.

Thank you for the comment, but we note that we collected the full range from 900 to 1700 nm because that is how the instrument collects data and is not programmable otherwise. There are pros and cons to the Nolin-Dozier method as well as other retrieval methods, including the spectrum fit method as you suggest. In addition to it being a well established method, we justify our selection of the Nolin-Dozier method over other methods, because it leverages an area of the spectrum with high signal to noise ratio for the instrument used in this study, and the shape of an ice absorption feature that is said to be less sensitive to the absolute magnitude of reflectance as a result of illuminations conditions. The full spectrum fit, however, is very susceptible to the absolute reflectance values and would equally weight NIR bands with high signal to SWIR bands

with low signal. Co-author Chris Donahue has experimented with this type of retrieval in a laboratory under idealized lighting conditions and has found spectrum fitting in the SWIR region to be problematic.

Line 212. "in the absence of ice absorption". I am not sure this is the best wording. Ice absorbs significantly at all NIR and SWIR wavelengths. You probably mean "if the 1030 nm band were removed"? Please clarify.

We appreciate your request for further clarification. We have reworded this phrasing to read: *"... represents the reflectance spectrum as if the ice absorption feature at 1030 nm were not present"*.

Line 215. How are your grain size results dependent on the SNICAR optical parameterizations, i.e., on the g and B values? You may compare the SNICAR values to those in Robledano et al. (2023) https://doi.org/10.1038/s41467-023-39671-3 and test whether the latest research would change your final results.

The results from the TARTES parameterization in Robledano et al. (2023) is unfortunately not directly translatable to the SNICAR model. Additionally, the Robledano et al. (2023) parameterization is for seasonal snow, where all snow samples examined were less than 400 kg/m$^3$, and are likely not suitable for firn. The SNICAR model represents grain shape using two alternative parameters known as shape factor and asymmetry parameter (He et al., 2017). There is currently an ongoing study to parameterize these shape parameters for SNICAR, similar to Robledano et al. (2023), and we are eager to see those results. We did, however, test the 4 selectable shapes within SNICAR that span a range of snow types and found that the spherical particle was the best.

Line 218 "Snow density negligibly affects snow reflectance". This is only true for a semi infinite homogeneous snow layer. In fact, it has no effect. Anyway, your sentence is correct in your context, since your core thickness is probably at least 3 times the e folding depth, but the statement is not generally true as suggested by your writing.

Given that density is an input to SNICAR, it is worthwhile to offer a simple and concise explanation as to why a constant value is justifiable.

Line 268 "by by"

Thanks for catching this typo. We have removed the duplicated word.

Figure 5a. The OIB flightlines appear black to me, not blue.

We now write "*navy blue*" in the figure caption. This may also appear clearer when the full image is uploaded for production.

Lines 371-372. Both Lehning 2002 and Vionnet 2012 have produced models for seasonal snow, not for firn. The physical processes determining grain growth are different. References for firn models would be more appropriate.

You are correct that these models were developed for seasonal snow. However, the SNOWPACK model is commonly used to simulate ice sheet firn (e.g., Groot Zwaaftink et al., 2013; Steger et al., 2017; Dunmire et al., 2021; Keenan et al., 2021; Thompson-Munson et al., 2023; Banwell et al., 2023). We cited Lehning et al. (2002) and Vionnet et al. (2012) because these publications present the traditional/optical grain size evolution descriptions used in the SNOWPACK model. We agree that there are likely issues in the simulated grain size evolution because the models were developed for seasonal snow, and we have had some initial conversations about this with researchers that contribute to the model development/applications. To make it more clear that many firn modeling studies rely on model parameterizations developed for seasonal snow, we have revised this section to read:

*"Traditional grain size measurements are often taken in the field, and these types of measurements have validated grain growth parameterizations (Lehning et al., 2002) in a physics-based land-surface snow model, SNOWPACK, that has been applied extensively to simulate the evolution of the firn layer on ice sheets and ice shelves (e.g., Groot Zwaaftink et al., 2013; Steger et al., 2017; Dunmire et al., 2021; Keenan et al., 2021; Thompson-Munson et al., 2023; Banwell et al., 2023). Furthermore, SNOWPACK also evolves effective grain size based on the description by Vionnet et al. (2012), but this parameterization is still dependent on empirical relationships with traditional seasonal snow grain size. Our effective grain size dataset provides a valuable opportunity to further investigate the discrepancies and the empirical relationship between traditional and effective grain size measurements of firn."*

Figure 6b. I really have trouble telling the purple from the black in the numbers at the top. More contrasted colors may be useful. I am not color-blind, by the way.

Because we no longer focus on the geometric correction in the main paper and exclusively focus our discussion on methodological differences driving the effective/traditional discrepancies, Figure 6 now looks like:

[Figure]

The panel originally in Figure 6b has been moved to the appendix where we discuss the geometric correction. We have updated the color scheme so that hopefully the different sections appear clearer.

[Figure]

Lines 379-383. This is not totally correct. You are mixing up snow and firn metamorphism, where processes are very different. In firn, the large radii of curvature mean that vapor diffusion is not always predominant, and surface diffusion can become important. Please see e.g., Maeno

and Ebinuma (1983) doi:10.1021/j100244a023. There are several other references on the topic. The temperature gradient in firn is too low to produce faceted crystals. The reference to Fierz 2009, which is for seasonal snow, is not adequate. This paragraph and the next one could probably be condensed to retain only the aspect actually relevant to your study. I suggest just focussing on firn data and processes. You may then, and separately, extend this to a snow discussion, but such a snow discussion does not seem useful to me. I however let you decide on this last point, but please remember that, for a scientific paper, the shorter, the better.

Thank you for raising this point. We agree that this could be written better. We have significantly revised this section of text, and we now focus more on firn processes and how firn grain shape and bonding results in observed discrepancies between effective/traditional grain sizes. We do, however, mention the differences between snow and firn grains at the end of the paragraph, since we believe that this is an interesting result that to our knowledge has not been shown before. Given that seasonal snow models are applied to simulate ice sheet firn, we believe that the different relationship between effective and traditional grain sizes in snow and firn is important to note.

The revised section now reads:

*Effective and traditional grain sizes are not expected to be the same since they are based on different grain properties. Effective grain size retrievals based on NIR reflectance spectra rely on measurements of ice absorption and are ultimately a measurement of optical path length. Alternatively, traditional observations measure the cross-sectional extent of the grain. Previous limited studies report effective snow grain sizes to be ~ 2 – 20 times smaller than traditional grain radii (e.g., Painter et al., 2007; Langlois et al., 2010; Leppänen et al., 2015). However, we find that effective grain sizes are consistently larger than traditional grain size, while the magnitude of the effective-to-traditional radius differences are smaller than in snow. This suggests a different relationship between the two measurements in snow and firn.*

*The high bias in effective grain size retrievals compared to traditional observations of firn grains is likely caused by the methodological differences in measurement techniques compounded by the shapes of firn grains. While firn grains are typically spherical or spheroidal (Alley, 1997; Meussen et al., 1999), similar to the shape assumed to generate our grain size lookup table, sintering processes further reduce total surface area as firn grains become bonded together. In relatively low density firn (~350 – 550 kg m$^{-3}$), bonds between grains will form primarily from grain-to-grain vapor diffusion until the radii of curvature become large enough that surface diffusion becomes an important mechanism driving neck growth (Maeno and Ebinuma, 1983). The lower specific surface area of bonded grains, where the space between the grains has become ice-filled compared to two individual grains with an air-filled grain boundary, will cause greater NIR absorption and thus the retrieval of a larger sphere with the same surface-to-volume ratio (Wiscombe and Warren, 1980). The discrepancies between effective and traditional grain*

*sizes can be further increased when the firn cores are cut into half-round sections. It is highly unlikely that all firn grains are cut exactly through the center. Therefore, a traditional grain size measurement will calculate the diameter of a smaller cross-section than if the spherical grains were cut through the middle. Because NIR wavelengths penetrate to a depth of a few grain diameters, artifacts introduced by firn core cutting will not significantly influence the absorption and resulting grain size retrieval. Therefore, in firn, we largely attribute the >1 ratio of effective-to-traditional grain size to both increasing the retrieved effective grain radius by firn grain bonding that decreases specific surface area of the firn medium, and decreasing the traditional grain measurement by firn core cutting that exposes cross sectional areas smaller than perfect hemispheres. Our reported biases still suggest a distinctly different relationship than seen in seasonal snow grain size comparisons. In seasonal snowpacks, large temperature gradients promote the prevalence of kinetic growth forms with high surface area-to-volume ratios. The thicknesses of these grains can be 50 times smaller than their surface extents and these forms may also be hollow (e.g., Taillandier et al., 2007). The effective grain diameter of needles and plates are similar to their thicknesses (Mätzler, 1997), while a traditional measurement of their maximum extent would be much larger, which likely results in smaller effective grain sizes compared to traditional measurements.*

Lines 398-421. Honestly, I am not too thrilled by your explanation of the grain size difference, as mentioned above. First of all, is there any actual observational evidence of faceted forms in firn? I have not seen any but am open to evidence. The evidence you propose seems to be just a modeling choice that has no real basis, if I understand correctly. There is of course the obvious fact that traditional grain size uses a section where grains are not always cut in their center, therefore showing a smaller section. NIR reflectance on the other hand, penetrates several grains thick, so that it probes to a depth of several grains. As detailed in the start of my comments, I suggest you consider that NIR probes air-ice interfaces only, and does not see grain-grain boundaries, while traditional grain size does consider these boundaries. Since grain boundaries make up a significant fraction of a grain perimeter in snow, NIR will inevitably produce a negative artifact intrinsic to the method. If you agree to this, you may mention it in your conclusion. And by the way, I then think that translating NIR reflectance in terms of specific surface area (SSA) rather than grain size would be much more physically meaningful. Grain size implicitly implies that the whole perimeter of a grain is an ice-air interface, which is not true. SSA makes no assumption. The principle of the method you use to determine grain size was developed for snow, where most of the perimeters are indeed ice-air interfaces. Now that you are moving to firn, I extremely strongly recommend that you adjust your interpretation to the reality of that new medium.

*We appreciate your suggestion to revise this section. As discussed above, this section has been substantially revised to focus on methodological differences driving the traditional/effective grain size discrepancies.*

We do wish to note that our data come from the top 10 of the firn column, where the firn layer has not yet become isothermal. While seasonal temperature gradients will create vapor pressure gradients leading to vapor diffusion, firn temperature data from Humphrey et al. (2012), Charalampidis et al. (2016), and Harper et al. (2023) show that meltwater or rainwater infiltration can create very steep (> 10 C/m), localized, and deep (> 5 m) temperature gradients that can promote faceting. Furthermore, observations of depth hoar layers in firn by Benson (1962), Alley (1988), Satow and Wantanabe (1990), and McDowell et al. (2023) provide additional evidence that faceting can occur in firn.

We still believe that the high level of similarity between our observations and the idealized grain shape proposed by Humphrey et al. (2021) is interesting, and both reviewers actively engaged with this section in their reviews. Because we agree that the shape was somewhat arbitrarily-chosen and highly idealized, we have moved this short discussion to the appendix. This shortens the main text of the manuscript, yet still provides some discussion with which future work can engage. In the main text of the manuscript at the end of this section, we now write:

*"While we largely attribute the discrepancies between effective and traditional firn grain size to the fact that NIR reflectance is governed by ice absorption and traditional grain size is controlled by observable cross-sectional area, we notice that a simple geometric correction relating the grain shape proposed in the grain scale model of meltwater movement through firn by Humphrey et al. (2021) can explain a majority of the effective-to-traditional grain size differences in our dataset. Because the truncated octahedron grain shape proposed by Humphrey et al. (2021) assumes a highly-idealized firn grain, we dedicate a discussion developing a geometric-based correction factor relating effective and traditional grain size in Appendix A."*

We add some justification of examining forms with hexagonal facets in the appendix and write:

*"To examine whether effective and traditional grain sizes in the firn cores can be explained by the geometric treatment of effective grains as spheres, we assume firn grains take the shape of truncated octahedra proposed by Humphrey et al. (2021) in their grain-scale model of meltwater movement through firn. While highly idealized, this shape treats firn grains as semi-rounded forms containing hexagonal faces. While deep firn is isolated from large temperature-driven vapor pressure gradients, temperature data from the shallow (< 10 m) firn column from Humphrey et al. (2012), Charalampidis et al. (2016), and Harper et al. (2023) show that meltwater/rainwater infiltration and refreezing can create steep temperature gradients conducive to facet formation below the depth of large seasonal temperature gradients."*

Line 415. "The the"

Fixed. Thanks again for catching the typo.

Line 437. I am not sure Fierz 2009 applies here.

Good point. We now say *"... according to the classification scheme of Benson (1962)..."*

Lines 438-441. Would not this insertion be better placed together with the previous insertion? Your decision.

Thanks for the good suggestion to move the text. We have replaced the insertion we made on line 432-433 with this section since it essentially duplicated the meaning.

Figure A4. Why use the mean annual temperature? Would not the summer temperature be more appropriate? Or even just the July-August mean temperature? Only periods when melting may take place are relevant, it seems.

This is a fair point. We have changed the shading to the summer (June, July, August) average temperatures.

[Figure]

Line 520. Should you refer the reader to Figure 5 here? Your call.

We now include a reference to Figure 5.

Lines 541-542. Are units correct here? Or m rather than mm? No dot please, it is a unit.

Thank you for catching this oversight. The units are in m w.e. yr$^{-1}$. We have made these changes and removed the periods.

Lines 534-545. I am not really convinced by your explanation of the absence of a melt layer in core 11. Do you mean that the high accumulation rate would have decreased the temperature gradient, which would have reduced surface grain growth and therefore increased albedo? This dos not seem to be written very clearly. Other factors would then come into play. For example, more light snow of lower thermal conductivity will reduce downward heat loss during the warm spell and lead to greater heating of surface snow. This effect would be opposite to the one you propose. A quantitative assessment including all energy-relevant processes would be required to reach the conclusion you propose, if I understand it correctly. How about that the different snow structure, likely lower density, would have favored preferential flow rather than matrix flow, so that some areas would be minimally affected by wetting? Core 11 could then have been drilled in a little-affected spot. Just a thought, your decision. Your subsequent paragraph in fact almost leads to this same suggestion.

Thank you for raising this point. We agree that the inclusion of a hypothetical albedo feedback at the end of this paragraph was a bit lazily pitched. We have removed the final sentence from this paragraph, so that the focus is just on Core 11 being in a different climate setting from the other cores, which could be why the 2012 melt signal is not evident in this core.

**References cited in our responses:**

Alley, R. B. (1988). Concerning the deposition and diagenesis of strata in polar firn. *Journal of Glaciology*, *34*(118), 283-290.

Banwell, A. F., Wever, N., Dunmire, D., & Picard, G. (2023). Quantifying Antarctic‑wide ice‑shelf surface melt volume using microwave and firn model data: 1980 to 2021. *Geophysical Research Letters*, *50*(12), e2023GL102744.

Benson, C. S. (1962). *Stratigraphic studies in the snow and firn of the Greenland ice sheet* (Doctoral dissertation, California Institute of Technology).

Bohn, N., Painter, T. H., Thompson, D. R., Carmon, N., Susiluoto, J., Turmon, M. J., ... & Guanter, L. (2021). Optimal estimation of snow and ice surface parameters from imaging spectroscopy measurements. *Remote Sensing of Environment*, *264*, 112613.

Charalampidis, C., Van As, D., Colgan, W. T., Fausto, R. S., Macferrin, M., & Machguth, H. (2016). Thermal tracing of retained meltwater in the lower accumulation area of the Southwestern Greenland ice sheet. *Annals of Glaciology*, *57*(72), 1-10.

Cook, J. M., Tedstone, A. J., Williamson, C., McCutcheon, J., Hodson, A. J., Dayal, A., ... & Tranter, M. (2020). Glacier algae accelerate melt rates on the south-western Greenland Ice Sheet. *The Cryosphere*, *14*(1), 309-330.

Dunmire, D., Banwell, A. F., Wever, N., Lenaerts, J., & Datta, R. T. (2021). Contrasting regional variability of buried meltwater extent over 2 years across the Greenland Ice Sheet. *The Cryosphere*, *15*(6), 2983-3005.

The Firn Symposium Team. (2024). Firn on ice sheets. *Nature Reviews Earth & Environment*, 1-21.

Groot Zwaaftink, C. D., Cagnati, A., Crepaz, A., Fierz, C., Macelloni, G., Valt, M., & Lehning, M. (2013). Event-driven deposition of snow on the Antarctic Plateau: analyzing field measurements with SNOWPACK. *The Cryosphere*, *7*(1), 333-347.

Harper, J., Saito, J., & Humphrey, N. (2023). Cold season rain event has impact on Greenland's firn layer comparable to entire summer melt season. *Geophysical Research Letters*, *50*(14), e2023GL103654.

Humphrey, N. F., Harper, J. T., & Pfeffer, W. T. (2012). Thermal tracking of meltwater retention in Greenland's accumulation area. *Journal of Geophysical Research: Earth Surface*, *117*(F1).

Keenan, E., Wever, N., Dattler, M., Lenaerts, J., Medley, B., Kuipers Munneke, P., & Reijmer, C. (2021). Physics-based SNOWPACK model improves representation of near-surface Antarctic snow and firn density. *The Cryosphere*, *15*(2), 1065-1085.

MacFerrin, M., Machguth, H., As, D. V., Charalampidis, C., Stevens, C. M., Heilig, A., ... & Abdalati, W. (2019). Rapid expansion of Greenland's low-permeability ice slabs. *Nature*, *573*(7774), 403-407.

McDowell, I. E., Keegan, K. M., Wever, N., Osterberg, E. C., Hawley, R. L., & Marshall, H. P. (2023). Firn Core Evidence of Two‑Way Feedback Mechanisms Between Meltwater Infiltration and Firn Microstructure From the Western Percolation Zone of the Greenland Ice Sheet. *Journal of Geophysical Research: Earth Surface*, *128*(2), e2022JF006752.

Negi, H. S., & Kokhanovsky, A. (2011). Retrieval of snow albedo and grain size using reflectance measurements in Himalayan basin. *The Cryosphere*, *5*(1), 203-217.

Robledano, A., Picard, G., Dumont, M., Flin, F., Arnaud, L., & Libois, Q. (2023). Unraveling the optical shape of snow. *Nature Communications*, *14*(1), 3955.

Satow, K., & Watanabe, O. (1990). Seasonal variation of oxygen isotopic composition of firn cores in the Antarctic ice sheet. *Annals of Glaciology*, *14*, 256-260.

Skiles, S. M., Donahue, C. P., Hunsaker, A. G., & Jacobs, J. M. (2023). UAV hyperspectral imaging for multiscale assessment of Landsat 9 snow grain size and albedo. *Frontiers in Remote Sensing*, *3*, 1038287.

Steger, C. R., Reijmer, C. H., Van Den Broeke, M. R., Wever, N., Forster, R. R., Koenig, L. S., ... & Noël, B. P. (2017). Firn meltwater retention on the Greenland ice sheet: A model comparison. *Frontiers in Earth Science*, *5*, 3.

Thompson-Munson, M., Wever, N., Stevens, C. M., Lenaerts, J. T., & Medley, B. (2023). An evaluation of a physics-based firn model and a semi-empirical firn model across the Greenland Ice Sheet (1980–2020). *The Cryosphere*, *17*(5).